# Structural basis for cofilin binding and actin filament disassembly

Kotaro Tanaka[1], Shuichi Takeda[1], Kaoru Mitsuoka[2], Toshiro Oda[3], Chieko Kimura-Sakiyama[1,5], Yuichiro Maéda[1,4] & Akihiro Narita[1]

Actin depolymerizing factor (ADF) and cofilin accelerate actin dynamics by severing and disassembling actin filaments. Here, we present the 3.8 Å resolution cryo-EM structure of cofilactin (cofilin-decorated actin filament). The actin subunit structure of cofilactin (C-form) is distinct from those of F-actin (F-form) and monomeric actin (G-form). During the transition between these three conformations, the inner domain of actin (subdomains 3 and 4) and the majority of subdomain 1 move as two separate rigid bodies. The cofilin–actin interface consists of three distinct parts. Based on the rigid body movements of actin and the three cofilin–actin interfaces, we propose models for the cooperative binding of cofilin to actin, preferential binding of cofilin to ADP-bound actin filaments and cofilin-mediated severing of actin filaments.

[1] Structural Biology Research Center, Graduate School of Science, Nagoya University, Furo-cho, Chikusa-ku, Nagoya 464-8601, Japan. [2] Research Center for Ultra-High Voltage Electron Microscopy, Osaka University, 7-1 Mihogaoka, Ibaraki, Osaka 567-0047, Japan. [3] Faculty of Health and Welfare, Tokai Gakuin University, Nakakirino-cyo 5-68, Kakamigahara, Gifu 504-8511, Japan. [4] Toyota Physical and Chemical Research Institute, 41-1, Yokomichi, Nagakute, Aichi 480-1192, Japan. [5] Present address: Department of Medical Life Systems, Faculty of Life and Medical Sciences, Doshisha University, 1-3 Miyakodani, Tatara, Kyotanabe, Kyoto 610-0394, Japan. Correspondence and requests for materials should be addressed to A.N. (email: narita.akihiro@f.mbox.nagoya-u.ac.jp)

Actin turnover enables the dynamic incorporation of force and structure into many cellular processes through polymerization and depolymerization of actin subunits. The actin-depolymerizing factor (ADF)/cofilin protein family consists of small actin-binding proteins of 13–19 kDa that play central roles in accelerating actin turnover by disassembling actin filaments (F-actin)[1].

ADF was originally identified in chicken embryo brain[2], whereas cofilin, which stands for "cofilamentous protein"[3], was isolated from porcine brain. Both proteins are categorized as members of the ADF/cofilin family based on homologous amino acid sequences, 3D structures and cellular functions. ADF and cofilin are also the smallest members of the ADF-H (ADF homology domain) protein family[4], a wider classification of proteins that includes multidomain proteins.

ADF/cofilins are expressed in all eukaryotes[5], and their function in accelerating actin turnover directly affects the dynamics of motile structures, such as listeria comet tails[6], lamellipodia[7], filopodia[8], and neural growth cones[9]. ADF/cofilins are also essential for the maintenance of contractile systems, including contractile rings[10], stress fibers[11], and muscles[12], through their regulation of actin filament quantity and/or length. The contributions of ADF/cofilins to cellular actin dynamics have broad-ranging implications in cancer cell invasion, nerve cell network construction, animal development, and many other biological functions[1].

ADF/cofilins bind to the side of the actin filament (F-actin) and preferentially interact with ADP-F-actin rather than ADP-Pi-F-actin or ATP-F-actin[13,14]. ADF/cofilin binding is cooperative[15,16-18] and forms an ADF/cofilin-binding cluster on the actin filament, which is saturated at a 1:1 ADF/cofilin:actin molar ratio[19].

After binding F-actin, ADF/cofilin severs the filament[7,14]. Severing mainly occurs at the pointed end (P-end) of an ADF/cofilin cluster[18,20]. ADF/cofilin can also accelerate depolymerization of the filament at basic pH (7.8 or 8.0) under certain conditions[18,21]. Depolymerization at the P-end is accelerated in an ADF/cofilin-saturated actin filament (hereafter referred to as a "cofilactin filament")[18]. When no ATP-G-actin is available, ADF/cofilin binds to and accelerates the dissociation of the last barbed end (B-end) subunit of a bare actin filament[18].

Following polymerization, actin filaments hydrolyze ATP, leaving the "old" filaments consisting of ADP-F-actin[22]. Preferential binding of ADF/cofilin to ADP-F-actin causes selective disassembly of the "aged" filaments. Thus, ADF/cofilin contributes to actin monomer recycling by accelerating actin filament turnover[7,13].

Actin forms a double–stranded filament[23]. Electron microscopy structural analyses of a cofilactin filament at low to medium resolution have revealed that one cofilin molecule binds to two adjacent actin subunits within a single strand and that the interaction shortens the helical pitch and weakens both the intrastand and interstrand contacts between the actin subunits[19,24]. However, because of the limited resolution of previous studies, the structural basis of the activity of ADF/cofilin on the actin filament remains unknown.

In the present study, via cryo-electron microscopy (cryo-EM) we obtain a 3.8 Å resolution cofilactin structure reconstructed from chicken cofilin and chicken skeletal muscle actin. The structure we obtained in this study enables us to determine the nature of the cofilin-binding-induced conformational transition in the actin subunit. During this transition, the inner domain (ID, subdomains 3 and 4) and subdomain 1 (SD1) of the actin subunit act as two independent rigid bodies. The transition, which is distinct from the G-F transition, is defined as a rotation of the SD1 around an axis relative to the ID. The present structure also allows us to propose models for how cofilin cooperatively binds the actin filament, how intra- and interstrand actin–actin contacts are weakened, and how cofilin severs the filament. Finally, the implications of the present structure are discussed in terms of how cofilin preferentially binds ADP-F-actin versus ATP-actin.

## Results

**Overall structure**. A cofilactin structure with a 3.8 Å resolution (Fig. 1 and Supplementary Fig. 1), reconstructed from chicken cofilin and chicken skeletal muscle actin, was determined by cryo-EM (Supplementary Table 3). Among the available ADF/cofilin isoforms, we chose chicken cofilin for the following reasons: (1) the biochemical properties of the recombinant protein are identical to those of tissue–purified cofilin[27], (2) mammalian cofilin has four cysteine residues partly exposed on its surface that tend to be oxidized to form dimers[28], whereas the two cysteine residues in chicken cofilin are buried within the protein, and (3) chicken cofilin forms stable cofilactin filaments at pH 6.6.

All of the main chains were traced without ambiguity (Fig. 1a, b) except for actin residues 1–6 (N-terminus), 41–49 (DNaseI-binding loop, D-loop) and cofilin residues 1–2 (N-terminus), indicating disorder or large fluctuations in these regions. Many side chains were observed directly (Fig. 1c, d). A single bound $Mg^{2+}$ ion and ADP (without density for a γ-phosphate) were clearly visualized (Fig. 1c and Supplementary Fig. 1c), indicating that cofilactin was bound to ADP and not to ATP under the experimental conditions. Cofilin interacts with two adjacent actin subunits in a single strand. The helical pitch was shortened ($\Delta\phi = 36°$, where $\Delta\phi$ is the helical rotation angle between two adjacent subunits on one strand) relative to the canonical actin filament ($\Delta\phi \sim 27°$), which is consistent with previous results[19,24]. However, owing to the improved resolution in this study, the main chain trace of the current model differed substantially from the structure previously reported at a 9 Å resolution (Supplementary Fig. 2a, b). The precise Cα positions enabled the characterization of the changes in the actin protomer structure that allow it to adopt the cofilactin structure. The distance between the center of gravity of one actin protomer and the filament axis (16.2 Å) is slightly larger than that of the F-actin (5JLF[25]) without residues 1–6 and 41–49, missing regions in cofilactin, for comparison (15.4 Å).

**Missing regions**. In the previous model[24], residues 41–49 of actin were modeled as a loop. In the present study, when a low-pass filter at 8 Å was applied to the present map, a similar density emerged at very low-density contour levels (Supplementary Fig. 2d). However, in the high-resolution map, the density resulting from residues 41–49 is absent (Supplementary Fig. 2c), indicating that this region is flexible. The flexibility of residues 41–49 is consistent with the fact that cleavage of actin between residues 47–48 by subtilisin is enhanced in cofilactin relative to F-actin[29] and the "stiffness cation" that binds to the missing region is released upon cofilin binding[30].

**Cofilin structure**. The overall cofilin structure is almost identical to the human cofilin-1 crystal structure, 4BEX[31] (Supplementary Fig. 2e, f). The largest difference between the present cofilactin structure and 4BEX is in the loop at residues 19–30 (Supplementary Fig. 2e, f), which includes the actin–cofilin interface at cofilin residues 18–23 (Supplementary Table 1). This difference is likely induced by actin-cofilin binding.

**Rotation between the outer and inner domains**. The relative orientations between the outer domain (OD, subdomains 1

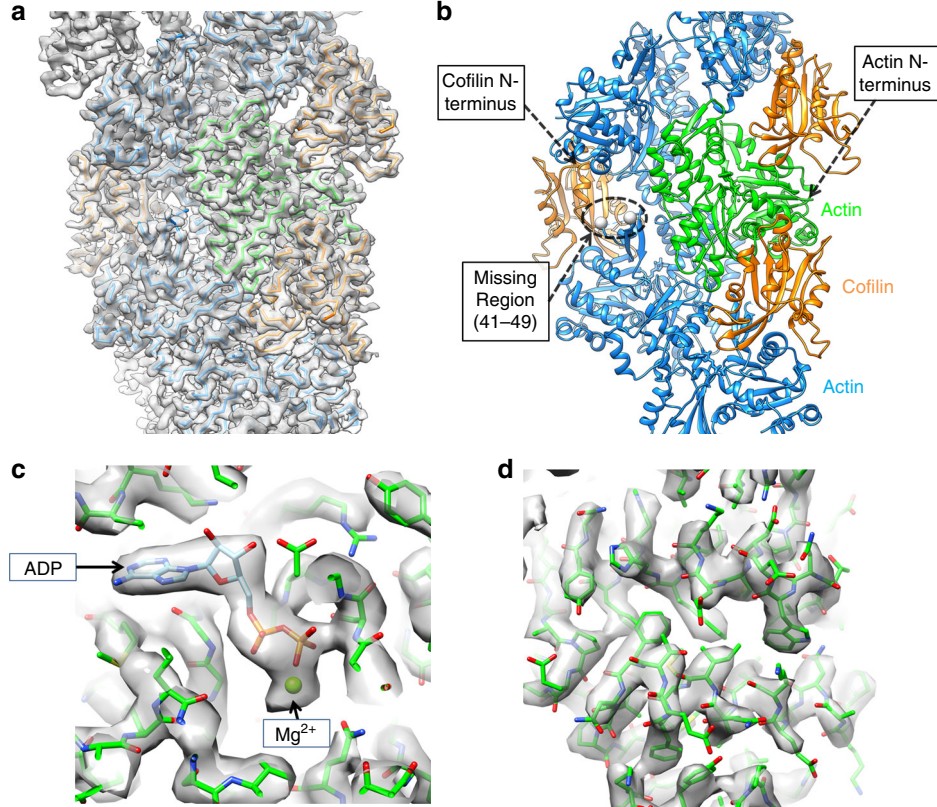

**Fig. 1** Overall structure of cofilactin. **a** Cryo-EM map at a 3.8 Å resolution. The main chains of the cofilactin model are superimposed. One actin subunit, the other actin subunits and cofilins are presented in green, cyan and orange, respectively. The P-end is at the top. **b** Ribbon diagram of the cofilactin model with the same color scheme as in **a**. The viewing angle is also identical to **a**. The position of one missing region (41–49 of actin) is indicated by a dotted ellipse. Residues 40 and 50 of actin, the boundaries of the missing region, are presented by white spheres in the dotted ellipse. The N-termini of actin and cofilin, where some residues are missing, are indicated by arrows. **c** Cryo-EM map around ADP. The ADP and $Mg^{2+}$ ions are indicated by arrows. **d** Cryo-EM map centered around actin residue 20

and 2) and ID (Fig. 2a) for monomeric actin (G-actin) and the actin filament (F-actin) differ significantly[23]. In the G-actin structure (G-form), the OD and ID are tilted with respect to each other. Upon polymerization, the OD rotates around the red axis in Fig. 2e and f and the molecule is flattened by reducing the tilting angle by 15.0° (F-form, Fig. 2c, d and Supplementary Movies 1, 2). We designated this rotational axis, which defines the center of the major conformational change in the G-F actin transition, as the "G/F axis". The structural change needed to form cofilactin has previously been proposed to be a rotation around the G/F axis[24]. However, our high-resolution density map clearly shows that the structural change between G-actin and cofilactin cannot be explained by a rotation around the G/F axis. Here, the actin protomer in the cofilactin structure is designated as "C-form" actin (Fig. 2a, b). The conformational transition from G-form to C-form actin is described by a 6.0° rotation of the OD relative to the ID around a new axis, designated as the "G/C axis" (orange in Fig. 2e, f and Supplementary Movies 3, 4), which is orthogonal to the G/F axis. This rotation causes the cleft between subdomains 2 and 4 (Fig. 2a) to become narrower, though the environment around ADP remains almost identical to that of G-actin (Supplementary Fig. 3e) because the nucleotide is positioned almost exactly on the G/C axis (Fig. 2e).

**Rigid bodies within the actin protomer.** We took a closer look at the structures of the ID and OD in isolation by comparing the typical actin structures available at atomic resolution. The present cofilactin structure and seven other actin structures were included

in this analysis: G-actin modified with TMR (1J6Z)[32]; G-actin bound to the tropomodulin actin-binding domain and gelsolin domain 1 (4PKH)[33]; actin bound to the twinfilin C-terminal domain (3DAW)[34]; the F-actin structure determined by cryo-EM (5JLF)[25]; the F-form actin structure from our recent crystal structure (Takeda et al., unpublished results); actin in the actin-profilin-VASP peptide complex (2PAV)[35]; and the actin portion of the actin-thymosin beta 4 hybrid protein (4PL7)[36]. All of the actins analyzed exhibit similar ID (Fig. 2g and Supplementary Fig. 3a, b) and SD1 (Fig. 2h and Supplementary Fig. 3c, d) structures. However, the relative orientations between ID and SD1 are diverse, suggesting the existence of two rigid bodies participating in conformational changes.

To more accurately define the rigid bodies, the actin structures previously listed were superimposed by aligning their regions of interest (ID or SD1). Residues with standard deviations in their Cα atom positions below 0.7 Å (see Methods for details) were considered to be fixed relative to each other during the structural transitions. The two rigid bodies were composed of residues making up most of the ID or SD1 region (Fig. 2i and Supplementary Table 2). This comparison process also identified two hinges between the two rigid bodies, located around residues 137 and 336, as the centers of the transitions among the actin forms (Supplementary Fig. 3a–d).

**Comparison with twinfilin bound actin.** Although many G-actin crystal structures have shown almost identical relative rigid body orientation[37], some actin crystal structures, including

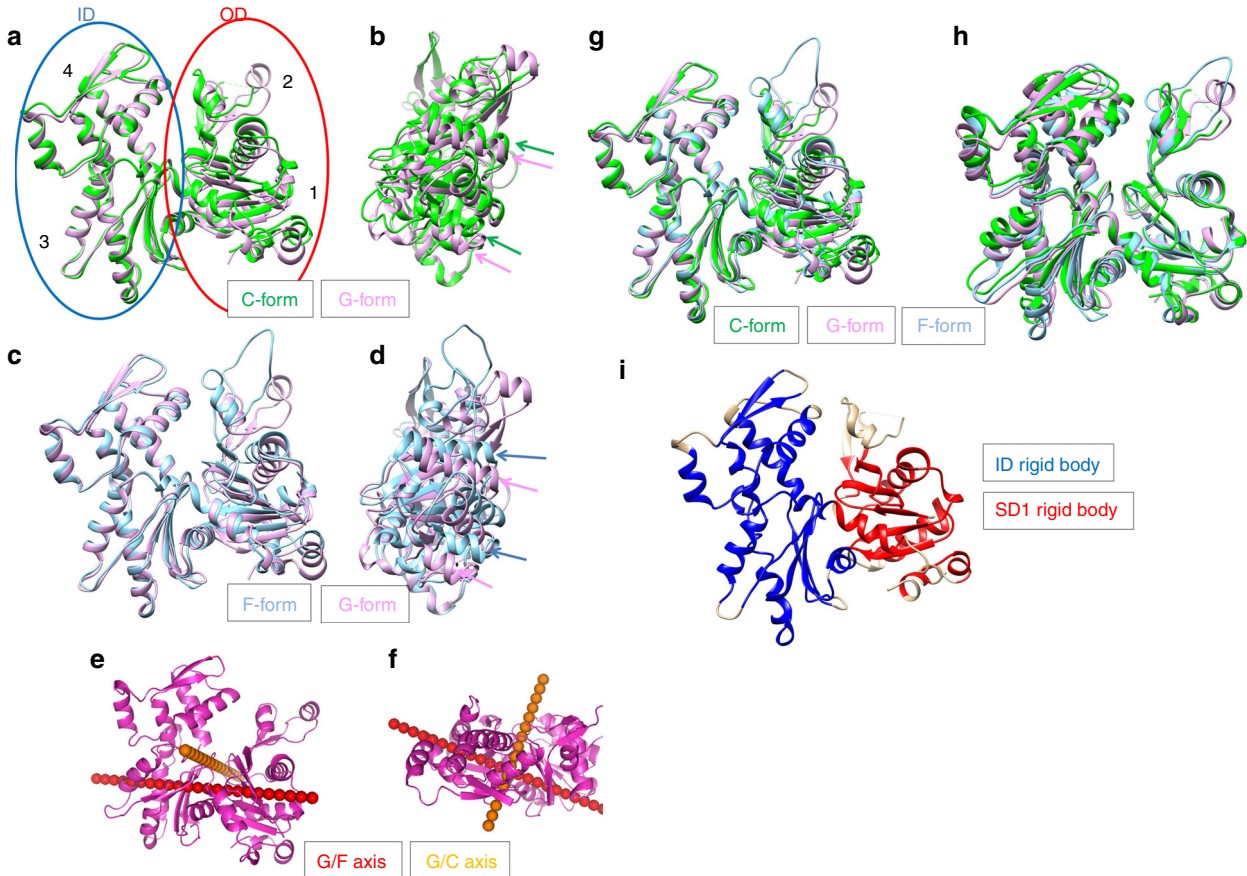

**Fig. 2** Actin structural changes in cofilactin. **a** Comparison between the G-actin (1J6Z, magenta, G-form) and actin subunit in cofilactin (green, C-form). The actin molecule is considered to have two domains, the inner domain (ID, blue ellipse) and outer domain (OD, red ellipse). Each domain is divided into two subdomains, OD into subdomains 1 and 2, and ID into 3 and 4. Each subdomain is indicated by a number (1, 2, 3 and 4). The structures are aligned through superimposition of the IDs. **b** Side view of **a**, rotated by 90°. Green and magenta arrows represent the orientation of two α-helices (79–95, 359–365) of the C-form and G-form, respectively. The two α-helices shift but do not tilt in the structural change. **c** Comparison between the G-form and F-form (5JLF, cyan). The structures are aligned through superimposition of the IDs. **d** Side view of **c**. Blue and magenta arrows represent the orientation of the two α-helices (79–95, 359–365) of the F-form and G-form, respectively. The two α-helices tilt in the structural change[23]. **e** Rotation axes. G/F axis (red) and G/C axis (orange) are superposed on the G-form (1J6Z, magenta). **f** Top view of **e**. **g** The G-form (1J6Z, magenta), F-form (5JLF, cyan) and C-form (green) are superposed by aligning the rigid bodies of the ID (blue in Fig. 2i). **h** The three forms are superposed by aligning the rigid bodies of SD1 (red in Fig. 2i) as in G. **i** Rigid bodies in the ID and SD1 are shown in blue and red, respectively

complexes with profilin[35], thymosin beta-4[36] and the C-terminal domain of twinfilin (twinfilinC)[34], differ substantially from the typical G-actin. TwinfilinC is the only ADF-H protein family member whose crystal structure has been revealed in complex with an actin monomer[34]. We found that the arrangement of the two rigid bodies in the twinfilinC:actin complex is very similar to that in cofilactin (Supplementary Fig. 3a, f–h). Therefore, the actin-twinfilinC structure (3DAW) represents the first published C-form structure. This result also suggests that the binding of ADF/cofilin to monomeric actin near the barbed face (subdomains 1 and 3, hereafter referred to as the "G-site") can induce the structural transition of actin into the C-form (Supplementary Fig. 3f–h).

**Actin–actin interactions**. Conformational changes in the actin subunit cause changes in actin–actin interactions within the filament, including intrastrand and interstrand contacts. In the actin filament, intrastrand interactions are stronger than interstrand interactions[23]. The intrastrand interactions in F-actin (Fig. 3a–c) can be divided into two categories, the ID–ID interaction between adjacent subunits and the specific interaction

between the OD of the lower barbed end subunit (B-subunit) and ID of the upper pointed end subunit (P-subunit)[25]. Only a few weak interactions between the ODs of the two subunits were observed. The structure of the ID–ID interface is essentially identical in cofilactin and F-actin (blue in Fig. 3a, b); however, the OD–ID interactions are almost lost in cofilactin because of the conformational change in the actin subunit structure (red in Fig. 3a, b). Only Arg62 contributes to the OD–ID interactions in cofilactin (yellow in Fig. 3c)

The number of interstrand interactions is also decreased in the cofilactin filament relative to F-actin (Fig. 3d, e and Supplementary Movies 5, 6).

**Actin–cofilin interactions**. Actin–cofilin interactions can be divided into three categories based on the definition of the rigid bodies (Supplementary Table 2), which are designated here as "F-site", "Gi-site", and "Go-site" (Figs. 4, 5b, Supplementary Table 1 and Supplementary Movie 7). The F-site, which was named by Mannherz et al.[38] and is known to be the F-actin binding site, is the interface between the bound cofilin and the B-subunit. The F-site on the B-subunit consists of residues in the

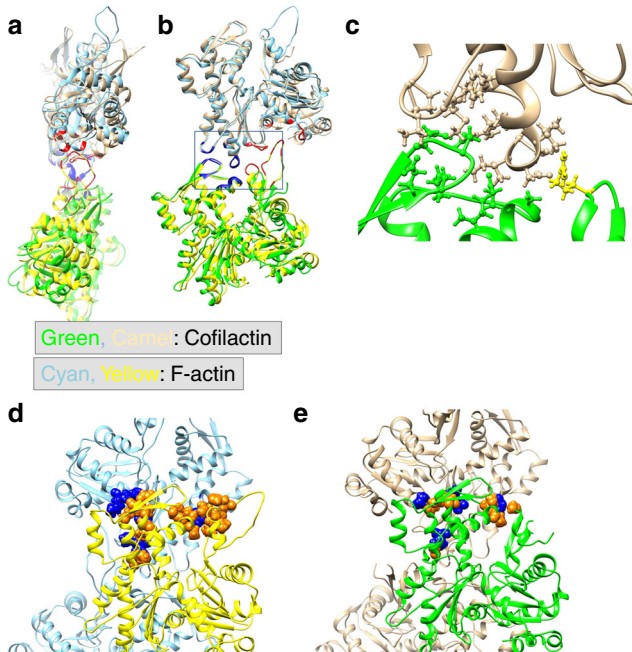

**Fig. 3** Actin–actin interactions. The P-end is at the top. **a** Comparison of the intrastrand interactions between F-actin (5JLF, yellow and cyan) and cofilactin (green and camel). The structures are aligned through superimposition of the IDs. The residues that are found in the interface of both F-actin and cofilactin are colored blue, while those found solely in F-actin are colored red. **b** A 90° rotation of **a**. **c** An enlargement of the intrastrand interactions in cofilactin in the rectangle in **b**. The side chains on the interface are shown as a ball and stick model. Arg62, the only residue of the outer domain contributing to the intrastrand interactions, is presented in yellow. **d** Interstrand interactions of F-actin. The interface residues, whose side chains form the interface, are shown as a space-filling model. The interface residues of the yellow subunit are shown in orange, and the other residues are shown in blue. **e** Interstrand interactions of cofilactin. The interface residues in a space-filling model of the green subunit are shown in orange and the other residues are shown in blue

SD1 rigid body with two additional residues, 57 and 336 (Supplementary Table 2) from subdomains 2 and 3, respectively. The standard deviation of residues 57 and 336 in the SD1 rigid body determination was 0.72 and 0.62 Å, respectively, which is on the threshold, indicating that the two residues move together with the SD1 rigid body. Therefore, the F-site interface on the actin subunit is located on the SD1 rigid body. As the rigid body structure is almost identical in all conformations, this result indicates that cofilin can bind to any form of actin through the F-site, provided there is no hindrance from another part of the structure. Cross-linking experiments have shown that binding through the F-site is also possible for G-actin, though the reported interaction was weak[38]. Furthermore, a model can be constructed of cofilin binding to F-actin through the F-site without any collision between the bound cofilin and F-actin (Figs. 4g, 5c), which suggests that cofilin can bind to F-actin through the F-site without any structural changes to actin. The K96Q mutant of human cofilin-1 inhibited the binding of cofilin to F-actin[39]. K96 is in the F-site and is preserved in chicken cofilin. The side chains of the residues surrounding K96 form a tightly packed actin-binding interface (Supplementary Fig. 4), consistent with the mutation having a drastic deleterious effect on cofilin-F-actin interactions.

The G-site, also named by Mannherz et al.[38], is the G-actin binding site and consists of the interface between the bound cofilin and the P-subunit. The G-site can be divided into two parts: the first, named the Gi-site, is located mainly on the ID rigid body (yellow and brown in Figs. 4a–f and 5b), and the second, named the Go-site, is located mainly on the SD1 rigid body (red in Figs. 4a–f and 5b). Because the relative positions of the rigid bodies differ between the F-form and C-form, G-site interaction requires structural changes in the actin subunit. The innermost area of the Gi-site (brown in Figs. 4 and 5b), named here the Gi_l-site, consists of four salt bridges or hydrogen bonds formed by extended side chains (Fig. 4d, e and Supplementary Table 1) with a distance of over 10 Å between the Cαs of each pair contributing to this interaction. A mutation of one of the residues in the Gi_l-site affects cofilin function[12], which supports the idea that interactions through the Gi_l-site play a significant role in the cofilin–actin interaction. We named the second part of the Gi-site the "Gi_s-site".

Actin residues from the Gi_s and Go-sites are tightly packed around the alpha helix-forming cofilin residues 111–119 (Fig. 4f), consistent with the fact that a cofilin peptide (residues 104–115) competes with cofilin in binding to actin[40]. Cofilin did not bind to the G146V mutant of *Dictyostelium* actin[41]. This result is also consistent with our model because G146, which is conserved in vertebrate skeletal actin, is located near the center of this tight interface.

When we constructed a model of cofilin bound to F-actin using the Gi-site interaction, substantial steric collisions occurred, mainly with the B-subunit (Fig. 4h). When cofilin bound to the Go-site, the collisions were even larger (Fig. 4i), indicating that cofilin cannot bind to either the Gi-site or the Go-site in an actin filament without conformational changes in the actin subunits.

## Discussion

The cryo-EM reconstruction revealed that the structure of cofilin-bound filaments is very similar to the actin-twinfilinC crystal structure (C-form), which is transitioned to by rotation around a newly defined rotational axis (G/C axis) between the domains. Furthermore, the actin protomer uses relative movements in two rigid bodies to switch between the forms. By fixing the relative orientation between the two rigid bodies, actin-binding proteins may control the conformation of the actin protomer. These findings provide important progress in understanding the polymorphism and structural dynamics of actin, in addition to understanding cofilin function.

When cofilin binds to the G-site of an actin subunit, it simultaneously interacts with the two rigid bodies (Fig. 5b), fixing their relative orientation in the C-form. Therefore, we propose that G-site binding is sufficient to induce the transition from the F-form to the C-form in the filament without F-site binding. The actin-twinfilinC structure strongly supports this hypothesis because twinfilinC fixes the actin monomer in the C-form by G-site binding (Supplementary Fig. 3f–h).

Profilin also induces conformational changes in the actin structure and binds to the two rigid bodies[35,36]. Cofilin, twinfilinC and profilin can be considered as examples of proteins that control the form of actin by fixing the two rigid bodies.

Cofilin binding induces a large structural change in the actin filament, and this transition is associated with the loss of some of the canonical interactions between the actin subunits in the filament. Therefore, the potential energy barrier for inducing the structural changes is likely to be significant. To reduce the height of the potential energy barrier in each step, we propose that more than two steps are required to elicit this structural change, a hypothesis that is supported by analysis of the kinetics of cofilin binding[42]. We propose a possible binding model based on our current structure and the cumulative results of previous research.

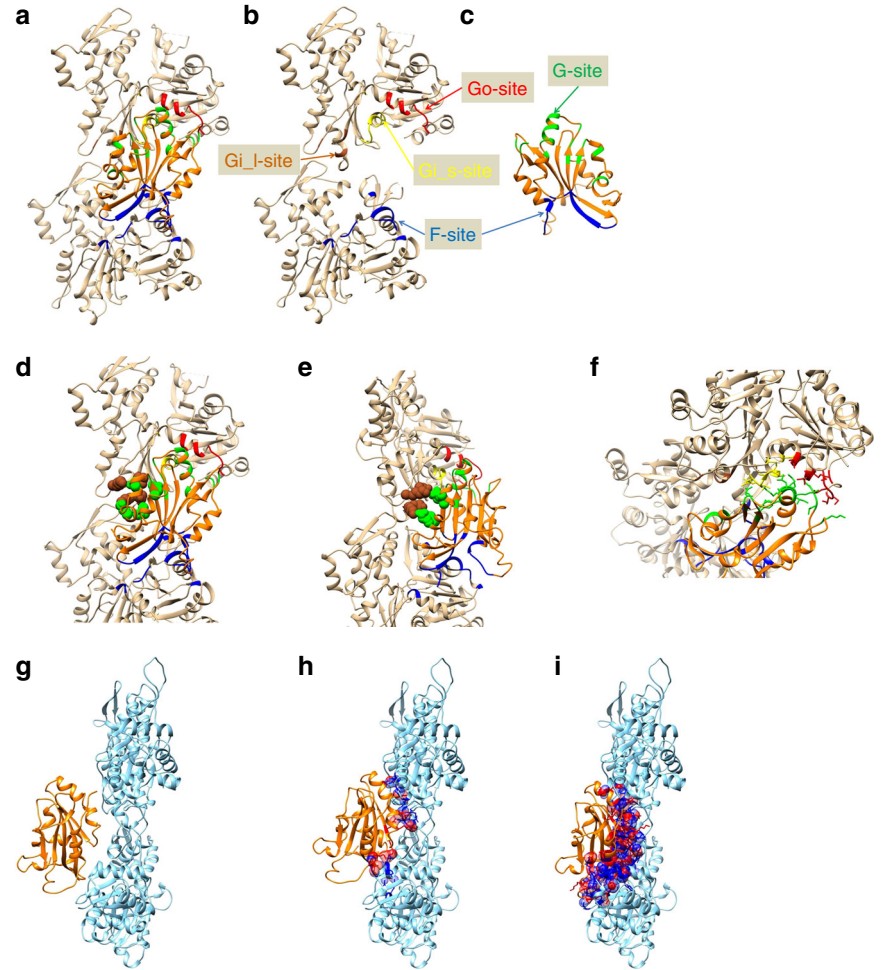

**Fig. 4** Actin–cofilin interactions. In **a**–**e** and **g**–**i**, the P-end is at the top. **a**–**f** Actin and cofilin are presented in camel and orange, respectively. The F-site (Cofilin-B-subunit interface), Go site (Cofilin-SD1 of the P-subunit interface), Gi_l site (Cofilin-ID of the P-subunit interface with long-range interactions) and Gi_s-sites (Cofilin-ID of the P-subunit interface which is not included in the Gi_l site) in the actin subunits are shown in blue, red, brown and yellow, respectively. In cofilin, residues forming the F-sites and G-sites are represented in blue and green, respectively. **a** Two actin subunits and the bound cofilin are presented. **b** Cofilin was removed from **a**. **c** Cofilin in **a** is rotated by 180°. **d** Residues contributing to the Gi_l-site are shown as a space-filling model. **e** A 50° rotation of **d**. **f** Residues within the Gi_s-sites and Go-sites are presented as sticks in an end-on view of the 111–119 helix of cofilin. **g**–**i** Models showing the steric hindrance expected when cofilin binds to F-actin. Colliding atoms on both actin and cofilin are represented as a space-filling model in blue and red, respectively. The other atoms of the residues containing the colliding atoms are represented as a stick model. **g** A model of cofilin bound to the actin filament through the F-site. No collisions occur. **h** A model of cofilin bound through the Gi-site. **i** A model of cofilin bound through the Go-site

We predict that the first step of cofilin binding to the actin filament is through F-site interactions of cofilin with the SD1 of an actin subunit because this interaction does not require a structural change in the actin subunit (Fig. 4g). However, as binding solely through the F-site is weak[38], the system is in equilibrium between the bound and unbound states (Fig. 5c, d).

The ADP-bound actin filament is less stable than the ATP or ADP-Pi-bound filament and is characterized by a higher critical concentration[22] and larger bending[43] and twisting[44] fluctuations. Computer simulations showed that an ADP-containing actin subunit undergoes larger domain fluctuations in the F- and G-forms[45] than an ATP-containing subunit, a result which suggests that in the "aged", ADP-bound actin filament, the orientation of SD1 relative to ID fluctuates.

In step 2, when the SD1 fluctuates, the cofilin bound through the F-site on the SD1 also moves and occasionally approaches the G-site on the P-subunit (Supplementary Movies 8, 9). In that position, the long side chains constituting the Gi_l-site of the P-subunit will interact with cofilin (Fig. 5e, and Supplementary Movies 8, 9) to form the Gi_l:cofilin interaction.

In step 3, cofilin is kept close to the G-site on the P-subunit, which increases the chances of inducing structural transition of the P-subunit to the C-form through full binding (Fig. 5f, and Supplementary Movies 8, 9). Simultaneously, the helical track linking two adjacent actin subunits becomes more twisted relative to the canonical right-handed helix (from $\Delta\phi = \sim 27°$ to $\Delta\phi = 35.8°$).

There is an alternative pathway (Fig. 5g, h) in which cofilin binds directly to the Gi_l-site without F-site binding when the B-subunit conformation resembles that at the end of step 2 due to structural fluctuations of the B-subunit (Fig. 5g). In this pathway, F-site binding is also important because the equilibrium between the bound and unbound states (Fig. 5c, d) keeps cofilin near the binding site.

This binding model accounts for the slow initial rate of cofilin's strong binding to F-actin (step 3 in our model), which is much slower than the collision limits[42,46]. Bound cofilin in step 1 must wait for the opportunity to interact with the G-site, which becomes possible through fluctuations of the actin subunit OD.

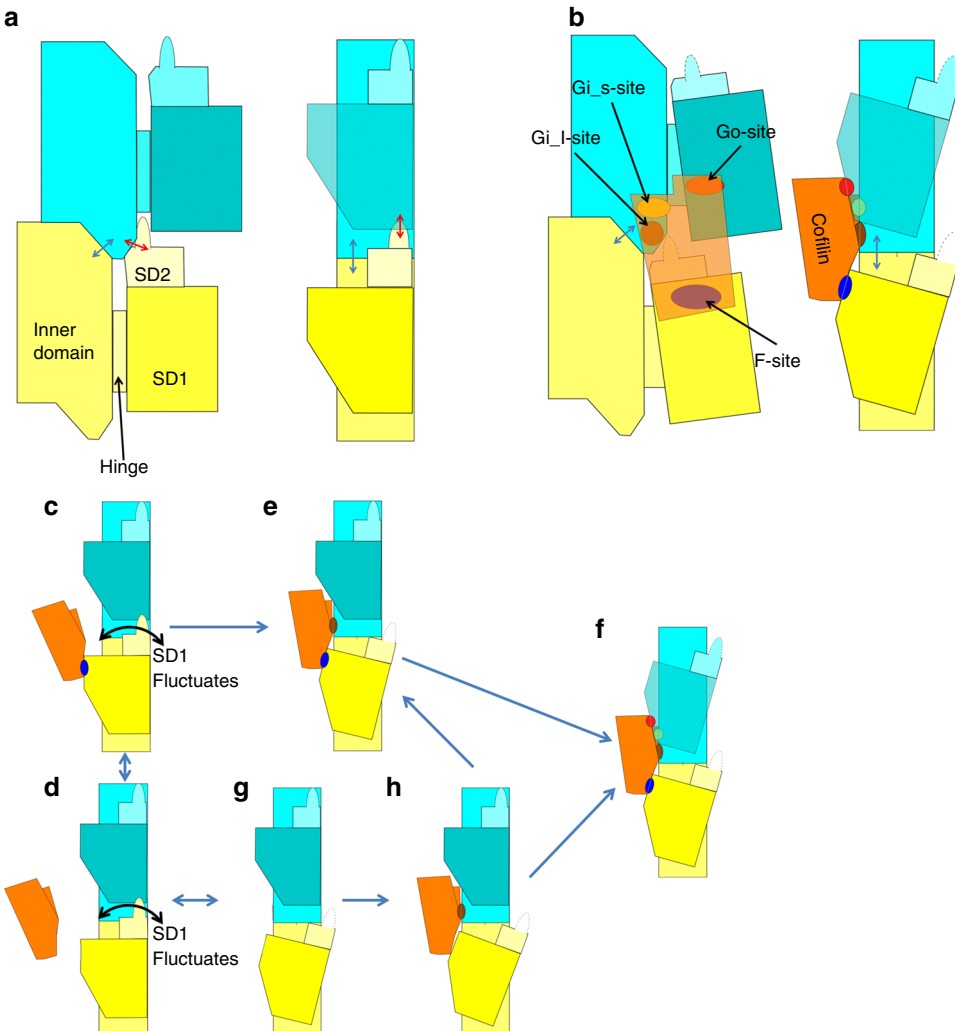

**Fig. 5** Schematic illustrations of intramolecular contacts. The P-end is at the top. Two actin subunits are shown (in yellow and cyan) from the longitudinal contact in one strand of the F-actin (**a**) and from the cofilactin filament (**b**). In **a** and **b**, the left panel is the front view and the right panel is the side view with the front surface on the left-hand side. In **c–h**, each panel is a side view. **a** A schematic illustration of F-actin. Each actin molecule is in the F-form, where the ID and OD are aligned to be flat. The ID–ID and OD–ID interactions are represented by blue and red arrows, respectively. **b** A schematic illustration of the cofilactin filament. Cofilin is in orange. Actin and cofilin interact with each other through three interaction sites, the F-site (blue), Gi-site and Go-site (red). The Gi-site is subdivided into two parts, the Gi_l-site (brown) and Gi_s-site (yellow). Part of subdomain 2 (residues 41–49) is disordered and presented by dotted lines. The outer domain is tilted and the nucleotide-binding cleft is closed. The ID–ID interactions remain (blue arrow) while the OD–ID interactions diminish (Fig. 3a, b). **c–h** Schematic illustrations of a multi-step model for cofilin binding to the actin filament. **c**, **d** Step 1. Cofilin binds to F-actin through the F-site (blue). The system at this stage is in equilibrium between the bound (**c**) and unbound (**d**) states. The OD of the ADP-bound actin subunit occasionally fluctuates (represented by the curved arrow). **e** Step 2. Cofilin bound through the F-site eventually approaches the Gi_l-site (brown), which is captured through binding of the flexible and extended side chains. **f** Step 3. The structural transition in the P-subunits is induced, and full G-site binding occurs (yellow and red). **g**, **h** An alternative pathway in which cofilin directly binds to F-actin through the Gi_l-site without F-site binding

This binding mechanism also explains why cofilin preferentially binds to ADP-F-actin. The potential barriers associated with step 1 (structural fluctuations) and step 3 (F-form to C-form transition) are smaller in ADP-F-actin compared with ATP-F-actin.

Cofilin binds to the actin filament cooperatively to form a cluster on the actin filament[17,18]. We propose a possible model for this process here. First, we discuss the cluster growth towards the P-end. We shall consider a situation where the first cofilin (Cof1) binds to two actin subunits (Ac1 and Ac2, both in the C-form) in the actin filament while the other actin subunits remain in the F-form. We term the P-end unbound actin subunit next to Ac2 "Ac3" and the B-end unbound actin subunit next to Ac1 "Ac0" (Fig. 6 and Supplementary Fig. 5a). Subsequently, another cofilin (Cof2) binds to the actin strand above the bound

cofilin through one of the three binding sites (F-site of Ac2, and Go-sites and Gi-sites of Ac3). Because of the Cof1 binding-induced conformational transition of Ac2 (Figs. 5 and 6b), in the Gi-site binding of Cof2 interacting with Ac3 (yellow and brown in Figs. 4a–f and 5) the number of colliding atoms between Cof2 and Ac2 is smaller than in de novo binding (Fig. 4h) by over 70%. A steric collision is expected with the flexible N-terminus of cofilin, whereas the other collisions are minor spatial overlaps of side chains. By contrast, binding via either of the other two sites would cause severe steric collisions (Supplementary Fig. 5b, c). This result indicates that Cof2 is likely to bind to the Gi-site of Ac3 with small rearrangements in the positions of the cofilin and actin subunit side chains (Supplementary Movies 8, 9 and Fig. 6c).

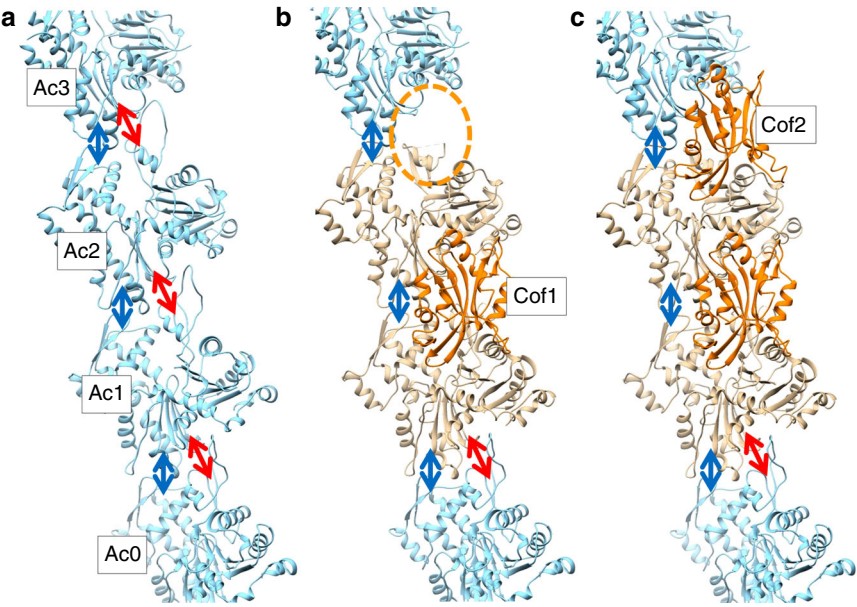

**Fig. 6** Model for cooperative binding of cofilin to the actin filament. The P-end is at the top. **a** One strand of F-actin. The ID–ID and OD–ID intrastrand interactions are presented by blue and red arrows, respectively. **b** Model of one cofilin (Cof1) binding to Ac1 and Ac2. Ac1 and Ac2 are replaced by the cofilactin structure by aligning the inner domains of the actin subunits. Because of the structural transition of Ac2, the OD–ID interaction between Ac2 and Ac3 is eliminated, creating a space (orange ellipse) for Gi-site binding of another cofilin (Cof2). The form of each actin subunit, C-form or F-form, is indicated in camel and cyan, respectively. At the boundary of Ac1/Ac0, the ID–ID and OD–ID intrastrand interactions between the actin subunits (Fig. 5a, blue and red arrows, respectively) remain because there was no significant structural change inside the ID of Ac1 (C-form) from the F-form and because Ac0 is in the F-form. **c** Model of Cof2 binding to the Gi-site of Ac3 in **b**

At this step, the helical twist between Ac2 and Ac3 is different from that of cofilactin, causing steric collisions when the F-site binding occurs (Supplementary Fig. 5b). However, the helical twist fluctuation should be enhanced because the OD–ID interaction between Ac2 and Ac3 is eliminated (Fig. 6b). Thus, F-site binding in combination with Gi-site binding will capture the twist in the cofilactin's one.

Here, the Gi-site binding of cofilin increases the chances of inducing structural transition of Ac3 to the C-form through full G-site binding without waiting for the structural fluctuation of Ac2 required for the de novo binding of cofilin to F-actin. Therefore, binding of an additional cofilin to the P-end direction of an existing cofilin cluster occurs faster than de novo binding.

Further, cluster growth towards the B-end would also be accelerated in this situation. Binding of the second cofilin (Cof2′) to the G-site of Ac1 would suffer from collisions like those resulting from de novo binding (Fig. 4h, i). Instead, Cof2′ could bind to the F-site of Ac0. For full binding, Cof2′ must wait for a G-site to approach through actin domain fluctuation. This situation is similar to that of the initial binding and limits the rate of cluster growth towards the B-end. However, when Cof2′ approaches the G-site, it can bind to the G-site without inducing a structural transition of Ac1, which is already in the C-form. Therefore, cluster growth at the B-end boundary should also be accelerated. Cof2′ binding to the G-site of Ac1 would induce transition of Ac0 into the C-form due to the requirement for the C-form of Ac0 for Cof2′ simultaneously binding to the G-sites and F-sites (Fig. 5b).

Thus, the acceleration mechanisms differ at the two ends of the actin filament, but we cannot determine based on the model whether cofilin cluster growth at the P-end boundary is faster than that at the B-end boundary[17,18].

Thus far, the discussion has only concerned a single strand of the actin filament. However, the actin filament consists of two strands, and the interactions between the two strands are also important.

Light microscopy observations have revealed that a substantial length of cofilin cluster is necessary for severing[20] the actin filament and that severing occurs at the P-end of the cofilin cluster. Wioland et al.[18] have also shown similar results, concluding that formation of a cofilin cluster is necessary for effective severing.

To form a cofilin cluster, cofilins must bind to both actin strands. We reason that it is unlikely that cofilin binding to one strand directly induces a structural transition in the opposite strand because the interstrand contact in cofilactin is weak (Fig. 3e). We propose a possible mechanistic model to address the questions of how cofilin binding to one strand propagates to the opposite strand and how the filament is severed, though our proposal contains speculations that need to be confirmed by future research.

This model is based on two assumptions: (1) in any actin-actin contact, any deviation from the canonical F-actin contact weakens the contact, and (2) severing is expected only at the point where the intrastrand contact is incomplete on both of the strands.

In Fig. 7b, one cofilin is bound to one strand of F-actin. Two cofilin-bound actin subunits in one strand change their positions and conformations. The positional shifts of the actin subunits could weaken the interstrand contacts, which is our first assumption. However, when we constructed hybrid actin filament models consisting of one cofilin-bound and one unbound strand, with the goal of observing collisions between the bound and the unbound strands (see Methods for details), almost no collisions were expected between the two strands (Supplementary Fig. 5e). Therefore, the opposite strand remains in the canonical F-actin structure (Fig. 7b).

The conformational transition of two actin subunits perturbs the intrastrand contact at the P-end boundary. There, the contact between the cofilin-bound C-form actin and the cofilin-free F-form actin within the same strand is partial, consisting of only an ID–ID contact, compared with the full ID–ID

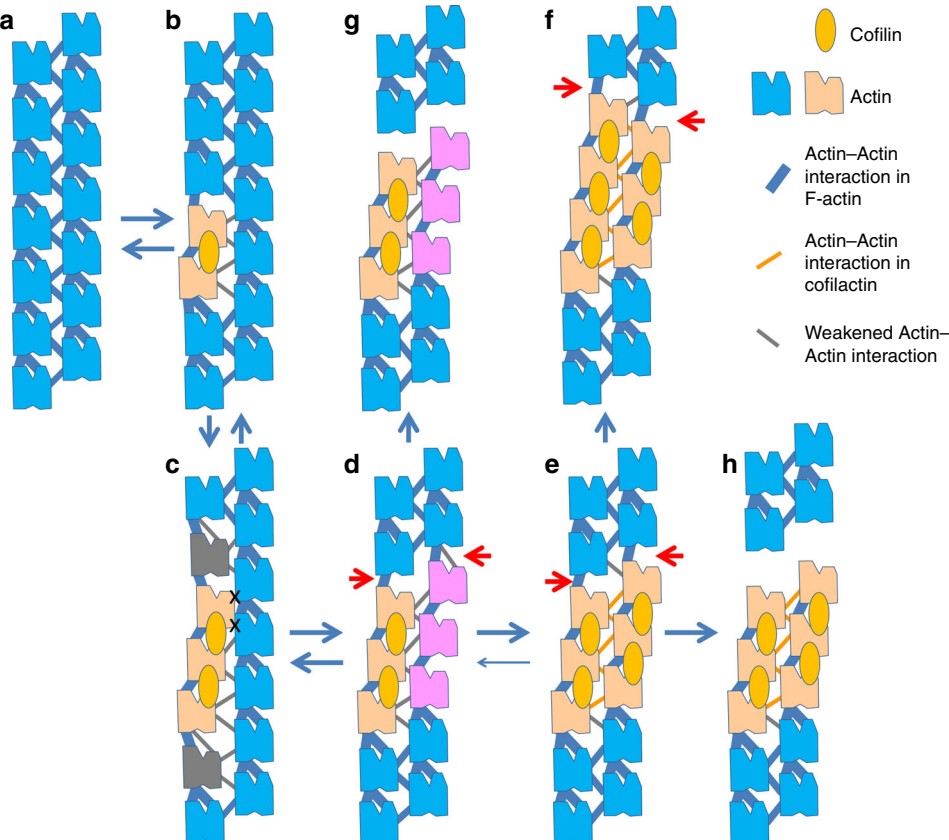

**Fig. 7** Model for severing. The P-end is at the top. **a** F-actin. The F-form actin subunits are represented in blue. The intrastrand actin-actin contact is represented by a pair of thick blue lines, one for the ID–ID interactions and the other for the OD–ID interactions. The interstrand actin-actin contacts are also represented by thick blue lines. **b** Upon binding of the first cofilin molecule (orange oval) to one strand of the F-actin, two actin subunits undergo conformational transition into the C-form (camel), which is associated with a twist change in the actin strand, and therefore changes in relative positions of subunits. We assume that these positional changes weaken the interstrand contacts, as indicated by the thin gray lines. **c** The second cofilin binds next to the first cofilin on the same actin strand. Collision occurs between the strands, as indicated by an "X". Accumulated position mismatches between actin subunits with and without bound cofilin may require adapter subunits at either boundary (gray). **d** As the size of the cofilin-bound section in the cofilin-bound strand increases, the helical twist transition propagates to the opposite strand. At this stage, the strain accumulated in the cofilin-bound strand and between the strands would be relieved. The cofilin-free actin subunits with the cofilactin helical twist are represented in magenta. The P-end boundary of the section is a candidate for severing (red arrows). **e** The helical twist change in the cofilin-free strand facilitates the binding of cofilin molecules to the strand, forming a double-stranded cofilin-bound cluster. The potential severing site remains (red arrows). Orange lines represent the interstrand interactions in cofilactin (Fig. 3e). **f** When another cofilin joins at the boundary of the cofilin cluster before severing, the cofilin cluster grows. The candidate severing site remains at the P-end boundary of the cluster (red arrowheads). **g, h** Severed filaments

contact plus OD–ID contact in the canonical F-actin strand (Figs. 3c, 5a, b, 6b). In contrast, at the B-end boundary, the intrastrand contact remains intact (Fig. 6b) because the contact interface of the upper subunit is almost exclusively at the ID (Fig. 3a, b), which undergoes no substantial structural changes.

At this step, no severing is expected because the cofilin-free strand remains intact.

When one or two additional cofilins bind longitudinally adjacent to the first bound cofilin, further conformational transition is induced together with a helical twist change, shifting the positions of the subunits in the cofilin-bound strand (Fig. 7c). The positional shifts have three consequences: mild collisions occur between the two strands (Supplementary Fig. 5f, g), the interstrand links are further weakened and the cofilin-free strand exhibits more structural fluctuations because of the weakened interstrand links. However, the cofilin-free sections of either strand also remain intact. Therefore, positional mismatches accumulate at the boundaries between the cofilin-bound and cofilin-free sections on the cofilin-bound strand, increasing strain energy in the actin subunits at both boundaries. This increase in

strain energy may transform the boundary subunits into unknown conformations as adaptor subunits (gray in Fig. 7c).

Based on the situation in Fig. 7c, the helical twist change induced by cofilin binding can propagate to the bare strand (Fig. 7d, magenta) by two factors: the enhanced structural fluctuations of the bare strand and the elimination of the accumulated strain at the sectional boundaries of the cofilin-bound strand (gray in Fig. 7c) when the helical twists of the two strands are compatible. The subunit-subunit interactions should be weakened in the bare strand with the cofilactin helical twist, which was our first assumption. At the P-end of this section, the intrastrand contact is only partial at either strand, making this point a good candidate for severing. In agreement with this conclusion, binding of a few cofilin molecules to the actin filament induces severing[15].

Figure 7e illustrates the formation of a cofilin cluster, where cofilin molecules bind to both strands. Based on the situation in Fig. 7d, cofilin molecules would bind to the cofilin-free strand due to the weakness of the intrastrand actin–actin contacts on the cofilin-free strand (magenta in Fig. 7d) relative to the canonical

ones, which reduce potential barriers for structural transition induced by cofilin binding.

At the P-end boundary of the cluster, the intrastrand OD–ID contact is eliminated from either strand, which should make the boundary a good candidate for location of severing (Fig. 7e). If cofilin binds to the boundary before severing occurs, the cofilin cluster grows and the severing point shifts towards the P-end (Fig. 7f).

The proposed model explains how cofilin severs the actin filament mainly at the P-end of a cofilin cluster[18,20]. It also explains why binding of only one cofilin molecule is not sufficient for effective severing[15,20].

A recent light microscopy study at a basic pH (7.8) revealed that ADF/cofilin can directly bind to the actin subunit at the B-end of the filament and accelerate depolymerization[18]. Furthermore, when the actin filament was fully saturated with ADF/cofilin, the depolymerization rates at the P-ends and B-ends were accelerated and decelerated, respectively. Consequently, the depolymerization rate of the fully decorated filament was comparable at both ends.

Our binding model is consistent with these findings. At the B-end, ADF/cofilin is accessible to the Gi-site of the B-end subunit with very minor steric hindrance (Supplementary Fig. 5d). Binding to the Gi-site induces the transition of the end-subunit into the C-form (Supplementary Fig. 6b), which weakens the subunit-subunit interactions and therefore accelerates dissociation of the end subunit from the filament.

The symmetrical depolymerization rates of the fully ADF/cofilin-saturated filament are also accounted for by our model. Without bound ADF/cofilin, the P-end subunit of the actin filament tilts toward and interacts with the opposite strand[37]. The extra interactions, which are specific to the P-end, slow depolymerization at this end because the P-end subunits need to dissociate from the filament to break this specific interaction (Supplementary Fig. 6a). When the actin filament is ADF/cofilin-saturated, the position of the tilted subunits at the P-end is rearranged, eliminating the extra interactions at the P-end. Therefore, the protein-protein contacts to be broken are the same in number and type for the dissociation of one actin subunit at either end (one actin-ADF/cofilin contact, one intrastrand actin-actin contact and one interstrand actin-actin contact, as illustrated in Supplementary Fig. 6c). Therefore, the depolymerization rate at either end should be comparable.

The models described here are based on earlier research and our present findings. Though they need to be confirmed or modified by future research, we believe that these models will be useful for the discussion of the ADF/cofilin function.

## Methods

**Protein preparation**. Actin was extracted from chicken skeletal muscle (purchased from a slaughterhouse as chicken meat) acetone powder, purified by polymerization and depolymerization steps[47], and further purified by gel filtration chromatography on a Sephacryl S-200 h column (GE Healthcare). Purified actin was stored on ice until use.

Chicken cofilin was expressed in *Escherichia coli* (BL21(DE3), Thermo Fisher) transformed with an expression vector of wild type chicken cofilin, kindly provided by H. Abe, Chiba University, Japan[27,48]. The *Escherichia coli* (*E. coli*) cells were harvested by centrifugation and then re-suspended in a buffer containing 50 mM Tris-HCl, pH 8.0, 20 mM EDTA, 1 mM DTT, and cOmplete Protease Inhibitor Cocktail (Roche). To lyse the cells, Bug Buster Protein Extract Reagent (Novagen) and 1 mg/ml lysozyme (from Egg White, Wako) were added to the cell suspension and incubated for 15 min, followed by sonication using the UD-201 sonicator (TOMY). The cell lysate was clarified by ultra-centrifugation at 165,000 × *g*. Cofilin was fractionated from the clarified cell lysate by ammonium sulfate fractionation (60–80% saturation) and purified by cation exchange chromatography using a HiTrapSP HP column (GE Healthcare) with a start buffer containing 20 mM MES (pH 6.2) and 5 mM EDTA and an elution buffer containing 20 mM MES (pH 6.2), 5 mM EDTA, and 1 M NaCl. The eluted sample was further purified by gel filtration chromatography on a Superdex 75 pg column (GE Healthcare) with a

buffer containing 5 mM PIPES (pH 6.6), 2 mM MgCl$_2$ and 50 mM KCl. After addition of 1 mM DTT, purified cofilin was frozen using liquid nitrogen and stored at –80 °C until use.

**Cryo-EM sample preparation**. Actin (12 μM) was polymerized in F-buffer (15 mM PIPES-NaOH (pH 6.6), 2 mM MgCl$_2$, 50 mM KCl, 0.1 mM EGTA, 0.2 mM ATP, and 1 mM DTT) for 90 min at room temperature. The acidic pH (6.6) stabilizes the cofilactin filament[49,50]. An equal volume of 50 μM cofilin in F-buffer without ATP was added to the F-actin solution and then incubated on ice for >30 min. The mixed solution (2.1–2.5 μL) was applied onto a grid with a holey carbon support film (Quantifoil R 1.2/1.3 Molybdenum) and plunge-frozen in liquid ethane by a Vitrobot Mark IV (FEI). Before use, the grids were cleaned by acetone, as previously described[51] then soaked in pure water overnight. Cleaned grids were glow-discharged with a 5 mA current for 60 s just before sample application. The parameters for plunge-freezing were as follows: blot time 2 s, humidity 100%, temperature 4 °C. After plunge-freezing, residual liquid ethane on the grid was thoroughly blotted off with filter paper. The best conditions for making cryo grids were screened by FEI Polara at Nagoya University.

**Image acquisition**. Samples were imaged using a Titan Krios microscope (FEI) operated at 300 kV installed with EPU software (FEI) at Osaka University. The imaging parameters were: nominal magnification 75,000, illuminated diameter 0.67 μm, objective aperture 100 μm, actual defocus 1.0–3.0 μm, dose rate 45 e$^-$/Å$^2$/s, exposure time 1 s, three image acquisitions per hole. Images were recorded with a FalconII detector (FEI) at a pixel size of 1.1 Å/pixel and a frame rate of 17 frames/s. For each image, the first two frames were discarded and the successive seven frames (accumulated dose of 24 e$^-$/Å$^2$) were saved and used for image processing (Supplementary Fig. 1a).

**Image processing**. A total of 1,111 images were collected in two microscope sessions (414 and 697 micrographs, respectively). Image processing was performed with RELION 2.0 software[52,53]. After motion correction with MotionCorr[54] and CTF estimation with Gctf[55], 1,093 images were selected for further image processing. Filaments were manually picked with e2helixboxer[56], after which 245,236 particles were extracted at a box size of 320 × 320 pixels. After 2D classification, 202,217 particles were selected. The initial 3D reference was prepared from the Galkin model of cofilin (PDB 3J0S)[24] with sxpdb2em[57]. The initial 3D reference was low-pass filtered to 25 Å, and 3D refinement was performed with a helical symmetry search without solvent flattening. Helical symmetry was converged to –162.1° twist/27.6 Å rise along the left-handed helix (corresponding to Δϕ = 35.8° in the main body of the text along the actin right-handed helical strand) and the resolution reached 4.3 Å. Additional 3D refinement was performed with fixed helical symmetry and solvent flattening, refining the resolution to 4.0 Å. RELION movie refinement, particle polishing and final 3D refinement were performed against the images collected in the second microscope session only (697 micrographs, 86,388 particles). The final resolution reached 3.8 Å (Supplementary Fig. 1b).

**Model building**. The initial atomic model consisted of five actin and three cofilin molecules that were prepared from an atomic model of G-actin (PDB 1J6Z)[32] and chicken cofilin (PDB 1TVJ). Actin was modified so that its N-terminus (residue 1–9), C-terminus (361–375) and subdomain 2 (36–67) were replaced with those of the atomic model of F-actin (PDB 2ZWH)[23]. Moreover, the ADP-associated Ca$^{2+}$ and methylated-histidine (residue 73) were replaced with Mg$^{2+}$ and unmodified histidine, respectively, and the N-terminal amine was acetylated. The initial atomic model was flexibly fitted to a cryo-EM density map with the cascade-MDFF protocol[58,59] on NAMD 2.12 software[60]. The calculation was performed under harmonic helical symmetry restraints[61] on all atoms except hydrogens, and the solvent effect was modeled with Generalized Born implicit solvent. After the cascade-MDFF calculation converged, Ramachandran outliers were fixed and refined with the phenix.real_space_refine program[62]. The resulting model was iteratively refined using COOT[63], MDFF and phenix.real_space_refine. After convergence, histidine 73 was replaced by a methylated-histidine.

**Rigid body search**. The IDs (residues 138–336) of the following eight actin structures were aligned with each other: G-actin modified with TMR (1J6Z), G-actin bound with a tropomodulin actin-binding and gelsolin domain 1 (4PKH), actin bound with twinfilin C-terminal domain (3DAW), F-actin structure determined by cryo-EM (5JLF), the F-form actin structure in our recent crystal structure (Takeda et al., unpublished results), actin in the actin-profilin-VASP peptide complex (2PAV), the actin portion of actin-thymosin beta 4 hybrid protein (4PL7) and the actin structure in the cofilactin. The boundary between the inner domain and subdomain 1 was assumed as the hinge position (Supplementary Fig. 3a–d). The standard deviation of each Cα position was calculated, and residues with <1 Å standard deviation in the inner domain were used as a putative rigid body. The eight actin structures were aligned again according to the putative rigid body, and the residues with <0.7 Å standard deviation were considered as the final rigid body of the ID. The rigid body of SD1 (7–35, 68–137, 337–372) was also determined by the same method.

**Rotation axis determination between two rigid domains**. The rotation axes of domain movement were obtained by the DomainSelect program (DynDom)[64] based on the two rigid bodies shown in Supplementary Table 2 and Fig. 2i.

**Modeling cofilins bound to the F-actin structure**. We constructed models of cofilins bound to the F-sites, Go-sites, and Gi-sites (Fig. 5b) on F-actin (Fig. 4g–i). For the F-site, we constructed the model in three steps. Step 1: one actin subunit and one cofilin bound to the actin subunit through the F-site were extracted from cofilactin (Supplementary Fig. 7a). Step 2: the SD1 rigid body (Fig. 2i) of the actin subunit in the extracted complex from Step 1 was aligned with the SD1 of an F-actin subunit (Supplementary Fig. 7b). Step 3: Cofilin in the aligned complex was combined with F-actin (Fig. 4g and Supplementary Fig. 7c).

For the Go-sites and Gi-sites, one actin subunit and one cofilin bound to the actin subunit through the G-site were extracted from cofilactin. The ID or SD1 rigid body of the extracted complex was aligned to an F-actin subunit, and cofilin in the aligned complex was combined with F-actin for the Gi-site or Go-site, respectively. We also made models for Fig. 6c and Supplementary Fig. 5a–d in the same manner.

**Constructing hybrid two-strand models**. We constructed hybrid actin filament models consisting of one cofilin-bound and one unbound strand to observe collisions between the bound and unbound strands (Supplementary Fig. 5e–h). We extracted a stretch of one strand from the cofilactin structure spanning two, three, four, and five actin subunits. In the F-actin structure, the corresponding part was replaced with the extracted cofilactin structure by aligning the inner domains of the center actin subunit with the counterpart of F-actin.

**Determining interfaces and steric collisions**. The distances between each atom in one molecule and the atoms in the interacting molecule were calculated. When the distance was less than the summation of the van der Waals radius×1.1, the atoms were considered as part of the interface between the two molecules. When the distance between non-hydrogen atoms was less than the summation of the van der Waals radius×0.5, the atoms were considered to collide. Colliding atoms existed only in the artificial models.

**Data availability**. The coordinates have been deposited in the Protein Data Bank (PDB) under accession code 5YU8, and the EM map has been deposited in the Electron Microscopy Data Bank (EMDB) (EMD-6844). All other data are available from the corresponding author upon reasonable request.

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

## Acknowledgements

This research was supported by JSPS KAKENHI (JP26251017 to Y.M. and A.N.; JP16H06280 to K.M.), JST PRESTO, Japan (10404 to A.N.), a Grant-in-Aid for JSPS Research Fellow (13J02335 to K.T.), Takeda Science Foundation, a Grant-in-Aid for Scientific Research on Innovative Areas—Platforms for Advanced Technologies and Research Resources "Advanced Bioimaging Support" and by Grants-in-Aid from "Nanotechnology Platform" (Project No. 12024046) of the Ministry of Education, Culture, Sports, Science and Technology (MEXT). We also acknowledge Prof. Hiroshi Abe of Chiba University, Japan, for advice on selecting cofilin species and providing expression systems. We thank Dr. Ryotaro Koike and Prof. Motonori Ota of Nagoya University, Japan, for sharing unpublished results concerning molecular dynamics simulations. We thank Prof. Robert Robinson (A*STAR and Okayama University) for polishing and editing a draft of this manuscript. We thank Prof. Koh Saito and Assoc. Prof. Makoto Kuwahara of Nagoya University, Japan, for granting us use of the FEI Polara, a cryo-EM. We also thank Michal Bell, PhD, from Edanz Group (www.edanzediting.com/ac) for editing a draft of this manuscript.

## Author contributions

K.T. and A.N. designed the research. K.T. performed most of the protein purification, electron microscopy, image analysis, and model building. K.M. performed the electron microscopy. C.S. performed the protein expression and purification. K.T., S.T., T.O., Y.M., and A.N. contributed to the structural interpretation. K.T., T.O., Y.M., and A.N. wrote the manuscript.

## Additional information

**Competing interests:** The authors declare no competing interests.

