## [Peer Review File · Nature Communications]

Reviewers' comments:

Reviewer #1 (Remarks to the Author):

The new structure described in this manuscript represents an important advance and deserves publication. The quality of the reconstruction from cryo-EM images is substantially better than previous EM reconstructions of cofilin bound to polymerized actin. The excellent density map shows many side chains and ADP clearly. Residues 41-49 were previously modeled as a loop and now appear disordered at higher resolution. This structure provides several important new insights regarding the binding of cofilin to actin filaments (similar to the actin-twinfilin crystal structure) and the effects of bound cofilin on the filament (including a second axis of rigid body rotation).

The authors propose plausible mechanisms for cooperative binding of cofilin and severing. However, the discussion over reaches and should be condensed and rethought in several ways.

Abstract and introduction

The authors assume that cofilins "disassemble actin" but this is not true for the whole family of these proteins. The best evidence for increasing the rate of depolymerization is for ADF at pH 7.8 or 8.0 (ref 17 and 20). This pH is not physiological. At pH 7 most cofilins that have been tested do not increase the rate of depolymerization beyond the rate of ADP-actin, including the cofilins tested in ref 17 and 20. Please do not propagate this misconception about cofilin in this paper. Severing is a more universal feature of these proteins.

Ref 19 rather than Ref 13 discovered the preference of these proteins for ADP-actin.

Figures

The figures can be improved in many ways. Several of the supplemental figures are more informative than the main text figures. Here are suggestions on the figures:

Fig. 1: I would label the ends of the disordered segments in at least one actin and one cofilin.

Fig. 2: The blue arrows in A are ambiguous and not helpful. The low contrast between the colors in A and B makes it difficult to appreciate the differences in the structures. The contrast is better in C and D. The arrows in B and D should be much smaller. Color-coded labels for the G-actin, F-actin and C-actin conformations in A-D and G-H and for the tilt axes in E and F would make it easier for the reader.

Fig. 3: These images show contacts between subunits in filaments \pm cofilin. The ribbon diagrams are good for orientation but not helpful for showing the actual interactions, which requires stick figures or space-filling models or a combination of those two styles. How were the structures aligned in A and B? The enlargement in C is not necessary, since readers can enlarge the image in B on their computer screens. The orientation of the images showing interstrand contacts in D and E is not helpful, since the contacts are difficult to see in such a projection image. I assume, but the legend does not explain that the contacts shown as space-filling are on the yellow subunit. The residues forming the other side of these contacts on the blue subunits are also of interest and not illustrated. The small number of blue contacts creates a different impression than the text, which states that the interstrand contacts are similar in filaments \pm cofilin.

Fig. 4: The legend does not define the F-, Go-, Gi₁- and Gi_s-sites, which are only explained in the main text. Please indicate the rotations with circular arrows on the figure. The following from the legend is unclear: "Colliding atoms on both actin and cofilin, are represented as a space-filling model in blue and red, respectively." I think that this means that an actin atom collides with cofilin is blue and a cofilin atom that collides with actin is red, but many red spheres are far from cofilin

and vice versa, so this is confusing. The lack of transparency in the rendering also eliminates from view all clashes beneath the proximal surface. Better rendering methods are available to show interpenetration of atoms in molecular models.

Fig. 5: One feature of the drawing is misleading. The text states "Part of subdomain 2 (residues 41–49) is disordered and presented by dotted lines." The line bordering the yellow projection is dotted, but the area is still filled with yellow implying that the structure is still there. In fact this whole region is disordered.

Fig. 6: I do not understand how there can be "no significant structural change inside the ID of the actin subunit in cofilactin" while the interstrand contacts change considerably when cofilin is bound (Fig. 3DE).

Fig. 7. Can probably be combined with Fig. 6.

Text

The explanation of the F-, Go-, Gi₁- and Gi_s-sites in the main text includes discussion of mutations, which is interesting, but would be more appropriate as part of the interpretation in the discussion. If that material is moved to the discussion, then the results section can get to the new observations more directly.

The discussion begins with "The cryo-EM reconstruction has revealed a new form of actin, the C-form," but the discussion goes on to state "the actin-twinfilinC structure strongly supports this hypothesis, since twinfilinC fixes the actin monomer in the C-form by G-site binding." Therefore, it would be more appropriate to say "The cryo-EM reconstruction has revealed that filaments with bound cofilin are very similar to the actin-twinfilinC crystal structure."

I am not sure about the journal's policies, but many journals would not approve of a discussion of unpublished work as presented on lines 275-287.

Line 293: The following seems misguided to me: "In step 3, the local concentration of cofilin close to the G-site on P-subunit is largely increased, which finally induces the structural transition of P-subunit to C-form through full binding." The local concentration is unlikely "to induce" a structural transition. More likely, the high local concentration simply allows cofilin to bind rapidly to the site and trap a high energy conformation of the actin, when rare thermal fluctuations open the site in the filament.

It seems to me that more than the new structure is required to support the three step binding mechanism shown in Fig. 5. The authors do not consider the lifetime of the weakly bound state, which is in rapid equilibrium with free cofilin. Depending on the relative rates of this exchange reaction and the transitions between conformations 1, 2 and 3, the pathway might have three steps as depicted, but free cofilin binding directly to filaments in the high energy state 3 might be the main pathway. Therefore, the discussion should explain what is known about the rates of all these reactions and how this supports the authors' hypothesis. Otherwise, the proposal should be clearly labeled as speculation.

Line 299: Ref 43 did not discover the very slow rate of binding of cofilin to actin filaments.

Lines 306-348: Cooperative binding. This has long been a fascination for the cofilin field and the new structure offers some insights. However, the mechanism proposed here involves considerable speculation. Coarse-grained MD simulations might test the validity of the hypothesis. Otherwise, the proposal should be condensed considerably and labeled as speculation.

Lines 349- 412: Severing. The authors offer a proposal for severing based on two assumptions. I

agree with the first but question the second, which is “Severing would be expected only at the point where the intra-strand contact is incomplete on both of the two strands.” This does not make sense. Both strands must contribute to the stability of the filament, so compromising either should make severing more likely. The loss of intrastrand contacts at each site of cofilin binding demonstrated clearly here for the first time must weaken the polymer and contribute to severing. The rest of the proposal is speculative. For example, the authors propose that “the opposite strand remains in the canonical F-actin structure (Fig. 7b)” when a single cofilin is bound to one strand. However, they only know the structure of filaments saturated with cofilin on both strands, so this assumption does not have experimental support in this paper. Can the authors marshal other evidence for the speculation? MD simulations might be helpful. Lines 392-395 propose another interesting mechanism, which is not supported by data in this paper but would be interesting to investigate by other methods. Lines 410-412 point out the importance of cofilin clusters for severing, but some cofilins sever at very low densities.

Lines 413-436: Depolymerization. The authors write “cofilin can directly bind to the actin subunit at the B-end of the filament and accelerates depolymerization (ref 17),” but I believe that ADF, not cofilin was used in the cited experiments, which were done at alkaline pH. The authors published important work on the structure of pointed ends, which may contribute to the rate of subunit dissociation. However, others showed that the crucial difference between the ends is their affinities for phosphate, so that P-ends have a higher probability to having bound ADP than barbed ends, which accounts for their different critical concentrations.

Reviewer #2 (Remarks to the Author):

In their manuscript ‘The structural basis of cofilin binding, severing and disassembling of actin filaments revealed by cryo-electron microscopy’, Tananka et al. present a 3.8 Å cryo-EM structure of the cofilin-F-actin complex.

The technical level of the study is high and the claimed resolution of the structure is justified.

Unfortunately, the structure provides only limited new biological insights in comparison to a structure of cofilin-F-actin at much lower resolution (Galkin et al. 2011). The model and extensive discussion are interesting and logical, but not based on novel findings. It could have been derived and written as review based on the 2011 paper of Galkin et al. The lack of novelty is also reflected in the way the manuscript is written. The results part that already contains extended discussion parts is shorter than the very long and detailed discussion.

Although the authors have a near-atomic resolution structure in hand, they do not describe the actin-cofilin interaction at ‘side chain level’. Was the resolution at the interface not high enough? Are the interactions of hydrophobic or electrostatic nature? Are there key residues involved? What can we learn from the structure that we did not know before?

The authors claim that their model of the cofilin-F-actin complex is substantially different from the previously reported structure (line 109). By looking at Supplementary Fig. 2, it becomes obvious that this is not the case. The orientation of the side chains might be different, but the authors do not write much about them in their manuscript.

The authors claim in their abstract that the ‘actin subunit structure in the cofilactin is distinct from those of F-actin and G-actin’. Here is a sentence of the abstract of the Galkin 2011 paper basically saying the same: ‘We show that.....is due to a unique conformation of the actin molecule unrelated to any previously observed state. The changes between the actin protomer in naked F-actin and in the actin-cofilin filament are greater than the conformational changes between G- and F-actin.’

The section 'Rigid bodies with the actin protomer' is unrelated to the primary data of the paper and should be removed or moved into the discussion part.

The section 'actin-cofilin interactions' should actually be the most important section of the manuscript describing the interaction of the two proteins in detail. It should be moved after the section 'cofilin structure'.

- Line 67 P-end should be defined.

- Line 101, 117 Residues 1-6, 41-49 should also be named as N-terminus and D-loop, respectively.

- Line 118 The map should be locally filtered according to the local resolution. This should reveal the density of the D-loop. This density should also be shown in a figure.

- The figures are in general very difficult to read, since many different structures or conformations are overlaid. The authors should present morphs between the different structures in supplementary movies.

- The figures are in general very 'naked' and can only be understood after careful reading of the legend. This should be changed and more details given directly in the figure using labels.

- Line 137ff: The conformational changes are difficult to see in the figure. I suggest to prepare a movie showing the transition of the single actin protomer changing its conformation from F-actin (F-form) to F-actin (C-form) and then from F-actin (C-form) to G-actin (G-form).

- Line 187ff: The ID-ID and OD-ID inter- and intra-strand interactions should be shown in detail – not just the backbone. Figure 3 does not help to understand this paragraph.

- Line 202: B-subunit should be explained.

- Movies: binding instead of binding or bining

- Line 274ff: In their discussion, the authors refer to crystal structures of actin in the F-form. The binding mechanism of cofilin is actually completely based on these structures and derived MD simulations. However, because the manuscript or the data have been not made available to the reviewers, it is difficult to judge on this part of the discussion. For example, is the move of SD1 really as extensive as described (line 289)?

- Figure 7 and 8. Actin protomers should not be shown as spheres but differently to show their polarity.

Reviewer #3 (Remarks to the Author):

This paper describes the structure of actin filament fully decorated with cofilin solved by cryo-electron microscopy and image analysis. The resolution is 3.8 Å, and this is high enough for the authors to build a reliable atomic model and discuss the binding interactions between actin protomers and cofilin as well as the conformational changes of actin upon cofilin binding in detail. Based on such structural information, the authors also extend discussion on how the structural changes of actin occur, why cofilin preferably binds to ADP-F-actin, how cooperative binding occurs along the actin filament, and how a clustered binding of cofilin severs the actin filament. This is a solid work, and the manuscript is comprehensible in most places.

Some points of concern are listed below for the improvement of the manuscript.

1. Many different aspects of the structure are nicely presented in the figures, but almost no labels of actin domains and residues make them rather difficult for general readers to understand what they see and even the direction of view of each figure.
2. The terms F-site, Gi-site (Gi_I-site, Gi_S-site) and Go-site are defined in the section of "Actin-cofilin interactions", but the reasons for these naming are not well explained. Since the F-site is on the P-subunit (actin subunit on the pointed-end side) and the G-site is on the B-subunit (actin subunit on the barbed-end side), it would be easier to follow the text if the F-site and G-site are termed P-site and B-site, respectively.
3. "The left side of the Gi-site" on lines 224-225 is difficult to follow in Fig. 4.
4. The last section of Discussion on "Effects on depolymerization rates at the ends" is a little too speculative and is better to be avoided at this stage.
5. Line 137: "this rotational axis as the G/F axis"
6. Line 282: "the network is less extended"
7. Lines 293 and 329-330: "local concentration" is an awkward wording.
8. Line 326: "the OD-ID interaction"
9. The sentence on lines 392-395 is difficult to follow.

Reviewer #1 (Remarks to the Author):

Comments:

The new structure described in this manuscript represents an important advance and deserves publication. The quality of the reconstruction from cryo-EM images is substantially better than previous EM reconstructions of cofilin bound to polymerized actin. The excellent density map shows many side chains and ADP clearly. Residues 41-49 were previously modeled as a loop and now appear disordered at higher resolution. This structure provides several important new insights regarding the binding of cofilin to actin filaments (similar to the actin-twinfilin crystal structure) and the effects of bound cofilin on the filament (including a second axis of rigid body rotation).

The authors propose plausible mechanisms for cooperative binding of cofilin and severing. However, the discussion over reaches and should be condensed and rethought in several ways.

Abstract and introduction

The authors assume that cofilins “disassemble actin” but this is not true for the whole family of these proteins. The best evidence for increasing the rate of depolymerization is for ADF at pH 7.8 or 8.0 (ref 17 and 20). This pH is not physiological. At pH 7 most cofilins that have been tested do not increase the rate of depolymerization beyond the rate of ADP-actin, including the cofilins tested in ref 17 and 20. Please do not propagate this misconception about cofilin in this paper. Severing is a more universal feature of these proteins.

Response:

I would like to thank referee 1 for the helpful suggestions to improve our MS. In the title, we used “severing” and “disassembling” simultaneously, giving the impression that disassembling does not include severing. Furthermore, it is true that acceleration of depolymerization was observed at high pH. We added a description to emphasize this (lines 65-66). However, we used the word “disassembling” for breaking the filament into small pieces, including severing and depolymerization in the Introduction. We believe that “disassembling” is better than “severing” for explaining

the ADF/cofilin functions because ADF/cofilin can really promote depolymerization, although it is unclear whether the depolymerization observed at high pH is important in the cell.

Finally, we deleted “severing” from the title.

Comments

Ref 19 rather than Ref 13 discovered the preference of these proteins for ADP-actin.

Response:

We cited Ref 19 (line 61).

Comments:

Figures

The figures can be improved in many ways. Several of the supplemental figures are more informative than the main text figures. Here are suggestions on the figures:

Fig. 1: I would label the ends of the disordered segments in at least one actin and one cofilin.

Response:

We followed the suggestion and labeled the missing regions (41-49 and N-terminus of actin, N-terminus of cofilin) in Figure 1b.

Comments:

Fig. 2: The blue arrows in A are ambiguous and not helpful. The low contrast between the colors in A and B makes it difficult to appreciate the differences in the structures. The contrast is better in C and D. The arrows in B and D should be much smaller. Color-coded labels for the G-actin, F-actin and C-actin conformations in A-D and G-H and for the tilt axes in E and F would make it easier for the reader.

Response:

We followed the reviewer’s suggestions. The color of the C-form was changed to green for a better contrast. The blue arrows were removed. We made the arrows in B and D smaller.

We also made movies to show the C-G and F-G conformational changes (Supplementary movies 1-4).

Comments

Fig. 3: These images show contacts between subunits in filaments \pm cofilin. The ribbon diagrams are good for orientation but not helpful for showing the actual interactions, which requires stick figures or space-filling models or a combination of those two styles. How were the structures aligned in A and B? The enlargement in C is not necessary, since readers can enlarge the image in B on their computer screens.

Response:

We replaced Fig. 3c and presented the interface in cofilactin by a ball and stick model. We also inserted a slightly more detailed description using this new presentation about the intrastrand contacts (lines 193-196).

The structures were aligned through superimposition of the IDs in Fig. 3, which we described in the legend.

Comments:

The orientation of the images showing interstrand contacts in D and E is not helpful, since the contacts are difficult to see in such a projection image. I assume, but the legend does not explain that the contacts shown as space-filling are on the yellow subunit. The residues forming the other side of these contacts on the blue subunits are also of interest and not illustrated. The small number of blue contacts creates a different impression than the text, which states that the interstrand contacts are similar in filaments \pm cofilin.

Response:

We agree with the reviewer, thus we changed the color coordinates. The new interface color is according to the subunit. Additionally, it is correct that the position of the interstrand contacts is not largely different. Only the number of contacts decreases in cofilactin. We changed the description to explain this (lines 197-198).

Comments:

Fig. 4: The legend does not define the F-, Go-, Gi₁- and Gi s-sites, which are only explained in the main text.

Response:

We added short descriptions to explain each site in the legend.

Comments:

The following from the legend is unclear: "Colliding atoms on both actin and cofilin, are represented as a space-filling model in blue and red, respectively." I think that this

means that an actin atom collides with cofilin is blue and a cofilin atom that collides with actin is red, but many red spheres are far from cofilin and vice versa, so this is confusing. The lack of transparency in the rendering also eliminates from view all clashes beneath the proximal surface. Better rendering methods are available to show interpenetration of atoms in molecular models.

Response:

We made the colliding atoms transparent in Fig. 4h and i. Moreover, the other atoms of the colliding residues are presented by a stick model, to show the connection between the main chain and the colliding atoms.

Comments:

Please indicate the rotations with circular arrows on the figure.

Response:

We added a description in Fig. 5.

Comments:

Fig. 5: One feature of the drawing is misleading. The text states “Part of subdomain 2 (residues 41–49) is disordered and presented by dotted lines.” The line bordering the yellow projection is dotted, but the area is still filled with yellow implying that the structure is still there. In fact this whole region is disordered.

Response:

We followed the reviewer’s suggestion and changed the colour of the region enclosed by the dotted line to white.

Comments:

Fig. 6: I do not understand how there can be “no significant structural change inside the ID of the actin subunit in cofilactin” while the interstrand contacts change considerably when cofilin is bound (Fig. 3DE).

Response:

We changed the description of the legend to explain this concept in detail.

Comments:

Fig. 7. Can probably be combined with Fig. 6.

Response:

We also thought this would be better. However, we could not do this because of the figure legend length limit stipulated by *Nature Communications*.

Comments:

Text

The explanation of the F-, Go-, Gi₁- and Gi_s-sites in the main text includes discussion of mutations, which is interesting, but would be more appropriate as part of the interpretation in the discussion. If that material is moved to the discussion, then the results section can get to the new observations more directly.

Response:

We agree with the reviewer. However, because of length limitations it was difficult to separate descriptions about previous mutant analyses and made an independent section in the Discussion .

Comments:

The discussion begins with “The cryo-EM reconstruction has revealed a new form of actin, the C-form,” but the discussion goes on to state “the actin-twinfilinC structure strongly supports this hypothesis, since twinfilinC fixes the actin monomer in the C-form by G-site binding.” Therefore, it would be more appropriate to say “The cryo-EM reconstruction has revealed that filaments with bound cofilin are very similar to the actin-twinfilinC crystal structure.”

Response:

We followed this suggestion.

Comments:

I am not sure about the journal’s policies, but many journals would not approve of a discussion of unpublished work as presented on lines 275-287.

Response:

We decided not to use unpublished data and we cited a study showing very similar computer simulation results to ours, although the editorial office did not comment about this.

Comments:

Line 293: The following seems misguided to me: “In step 3, the local concentration of cofilin close to the G-site on P-subunit is largely increased, which finally induces the

structural transition of P-subunit to C-form through full binding.” The local concentration is unlikely “to induce” a structural transition. More likely, the high local concentration simply allows cofilin to bind rapidly to the site and trap a high energy conformation of the actin, when rare thermal fluctuations open the site in the filament.

Response:

We agree with the referee. Thus, we changed the description (lines 293-295).

Comments:

It seems to me that more than the new structure is required to support the three step binding mechanism shown in Fig. 5. The authors do not consider the lifetime of the weakly bound state, which is in rapid equilibrium with free cofilin. Depending on the relative rates of this exchange reaction and the transitions between conformations 1, 2 and 3, the pathway might have three steps as depicted, but free cofilin binding directly to filaments in the high energy state 3 might be the main pathway. Therefore, the discussion should explain what is known about the rates of all these reactions and how this supports the authors’ hypothesis. Otherwise, the proposal should be clearly labeled as speculation.

Response:

The model presented here is one possible model, which requires confirmation by further experiments. We added a sentence emphasizing that it is a possible model (lines 447-450).

As indicated by the reviewer, another possible pathway is that cofilin binds directly to the Gi₁-site without F-site binding when the B-subunit conformation is similar to that at the end of step 2 (Fig. 5g) by the fluctuation of the B-subunit. We described this pathway in Fig. 5 and in the main text (lines 298-302).

Comments:

Line 299: Ref 43 did not discover the very slow rate of binding of cofilin to actin filaments.

Response:

Ref. 43 measured the slow binding rate of cofilin to an isolated site on the actin filament. It was also cited by De La Cruz et al., 2010, Biophysical Journal, as a paper showing the slow binding rate.

Comments:

Lines 306-348: Cooperative binding. This has long been a fascination for the cofilin

field and the new structure offers some insights. However, the mechanism proposed here involves considerable speculation. Coarse-grained MD simulations might test the validity of the hypothesis. Otherwise, the proposal should be condensed considerably and labeled as speculation.

Response:

The model presented here is also one possible model, which requires confirmation by experiments. We added a description to emphasize this (Lines 447-450).

Comments:

Lines 349- 412: Severing. The authors offer a proposal for severing based on two assumptions. I agree with the first but question the second, which is “Severing would be expected only at the point where the intra-strand contact is incomplete on both of the two strands.” This does not make sense. Both strands must contribute to the stability of the filament, so compromising either should make severing more likely. The loss of intrastrand contacts at each site of cofilin binding demonstrated clearly here for the first time must weaken the polymer and contribute to severing. The rest of the proposal is speculative. For example, the authors propose that “the opposite strand remains in the canonical F-actin structure (Fig. 7b)” when a single cofilin is bound to one strand. However, they only know the structure of filaments saturated with cofilin on both strands, so this assumption does not have experimental support in this paper. Can the authors marshal other evidence for the speculation? MD simulations might be helpful. Lines 392-395 propose another interesting mechanism, which is not supported by data in this paper but would be interesting to investigate by other methods. Lines 410-412 point out the importance of cofilin clusters for severing, but some cofilins sever at very low densities.

Response:

It is correct that the model is just one possible model. We stated this in the MS (lines 364-367, 447-450).

Light microscopy observations (Gressin et al., 2015, Current Biol.) have shown that severing by cofilin requires a substantial length of cofilin cluster (23 molecules) at a very low cofilin concentration (180-360 nM) at pH 7.0. Wioland et al., 2017, Current Biol. have also shown similar results at pH 7.8. Therefore, we believe a cluster containing more than two ADF/cofilin molecules is necessary for effective severing. To explain why a cluster of ADF/cofilin is necessary for severing, the second assumption is required.

Comments:

Lines 413-436: Depolymerization. The authors write “cofilin can directly bind to the actin subunit at the B-end of the filament and accelerates depolymerization (ref 17),” but I believe that ADF, not cofilin was used in the cited experiments, which were done at alkaline pH.

Response:

We added a description that the experiments were done at alkaline pH (line 424). Wioland et al., 2017, used mouse cofilin 1 and 2 in addition to ADF. Wioland et al. claimed that Cofilins also accelerated depolymerization although ADF was more effective. We replaced “cofilin” by “ADF/cofilin” in this section to make the descriptions more accurate.

Comments:

The authors published important work on the structure of pointed ends, which may contribute to the rate of subunit dissociation. However, others showed that the crucial difference between the ends is their affinities for phosphate, so that P-ends have a higher probability to having bound ADP than barbed ends, which accounts for their different critical concentrations.

Response:

It is correct that the affinities for phosphate are another important factor although the difference between the polymerization and depolymerization rates at both ends can induce different nucleotide states at both ends with ATP hydrolysis in the filament (Narita et al., 2011). However, describing this in detail is not constructive for this MS, hence we deleted the description about treadmilling.

Comments:

Reviewer #2 (Remarks to the Author):

In their manuscript ‘The structural basis of cofilin binding, severing and disassembling of actin filaments revealed by cryo-electron microscopy’, Tananka et al. present a 3.8 Å cryo-EM structure of the cofilin-F-actin complex.

The technical level of the study is high and the claimed resolution of the structure is justified.

Unfortunately, the structure provides only limited new biological insights in comparison to a structure of cofilin-F-actin at much lower resolution (Galkin et al. 2011). The model and extensive discussion are interesting and logical, but not based on novel findings. It could have been derived and written as review based on the 2011 paper of Galkin et al. The lack of novelty is also reflected in the way the manuscript is written. The results part that already contains extended discussion parts is shorter than the very long and detailed discussion.

Although the authors have a near-atomic resolution structure in hand, they do not describe the actin-cofilin interaction at ‘side chain level’. Was the resolution at the interface not high enough? Are the interactions of hydrophobic or electrostatic nature? Are there key residues involved? What can we learn from the structure that we did not know before?

The authors claim that their model of the cofilin-F-actin complex is substantially different from the previously reported structure (line 109). By looking at Supplementary Fig. 2, it becomes obvious that this is not the case. The orientation of the side chains might be different, but the authors do not write much about them in their manuscript.

Response:

We believe our MS provides a substantial amount of new data to better understand the ADF/cofilin functions. Furthermore, we proposed several models based on our new findings.

The rmsd between our cofilactin structure and 3j0s.pdb (Galkin et al., 2011) is 1.6 Å, which is not large considering the resolution of Galkin’s map. However, for

determination of the rigid bodies, it is too large because the threshold of deviation to determine the rigid body is 0.7 Å. We added a description to describe rmsd in the legend of Supplementary Fig. 2.

Based on the precise cofilactin structure, we found:

1. The precise actin structure in C-form.
2. Actin has two rigid bodies and its forms can be described by the relative position of the two bodies.
3. The actin-twinfilin structure was essentially the same as the C-form in terms of the two rigid bodies' orientation.
4. Regarding the actin-cofilin interaction, the F-site contains residues of the SD1 rigid body, while the G-site contains residues of the two actin rigid bodies.
5. Based on findings 2–4, it is plausible that cofilin binding to the G-site induces structural transition in the actin form.
6. The C-form actin loses the OD-ID interaction at the pointed end side, while the ID-ID interaction is the same as F-actin.

Findings 1-5 are completely new and more precise information about finding 6 could be achieved. We can propose models explaining cofilin functions based on these findings. Unfortunately, we could not find a small number of key residues for the interactions because many residues are included in the interface. The Gi_1-site consists of hydrogen bonds and electrostatic interactions as described in the text. The other sites are a mixture of hydrophobic, electrostatic and hydrogen bond interactions as usually found in protein-protein interactions. We described some of the interactions at the residue level, but we think that a detailed description of each interface is space consuming and not helpful for understanding the cofilin function. Readers can download the deposited map and pdb file, if they want to explore the interactions in more detail.

Comments:

The authors claim in their abstract that the 'actin subunit structure in the cofilactin is distinct from those of F-actin and G-actin'. Here is a sentence of the abstract of the Galkin 2011 paper basically saying the same: 'We show that.....is due to a unique conformation of the actin molecule unrelated to any previously observed state. The changes between the actin protomer in naked F-actin and in the actin-cofilin filament are greater than the conformational changes between G- and F-actin.'

Response:

It is true that Galkin et al. described that the actin form in the cofilactin is different from the F- or G- form. However, because of lack of the precise track of the main chain, they

interpreted that the structural change for forming cofilactin is a rotation around the G/F axis. In their model, the F-, G- and C- form could be described by different angles around the G/F axis. The C-form has a larger rotation angle than the G-form and they described it by “greater conformational change”. This is completely different from our current model. The rotation axis between the G-form and F-form is orthogonal to the rotation axis between the G-form and C-form.

Comments:

The section ‘Rigid bodies with the actin protomer’ is unrelated to the primary data of the paper and should be removed or moved into the discussion part.

Response:

The rigid body finding is one of the main findings in our MS and it is indispensable for understanding the cofilin functions. We believe it should be left in the Results section.

Comments:

The section ‘actin-cofilin interactions’ should actually be the most important section of the manuscript describing the interaction of the two proteins in detail. It should be moved after the section ‘cofilin structure’.

Response:

To understand the actin-cofilin interactions, the descriptions of the rigid bodies and actin-actin interaction are necessary. We would like to keep the current order.

Comments:

- Line 67 P-end should be defined.

Response:

We followed this suggestion (line 65).

Comments:

- Line 101, 117 Residues 1-6, 41-49 should also be named as N-terminus and D-loop, respectively.

Response:

We followed this suggestion.

Comments:

- Line 118 The map should be locally filtered according to the local resolution. This

should reveal the density of the D-loop. This density should also be shown in a figure.

Response:

It is not popular to locally filter the map by the local resolution because the absolute resolution value for each region is largely dependent on the parameters or programs used, although the local resolution is very useful to understand the gross tendency of heterogeneous fluctuations of the structure. Instead of showing a locally filtered map, we presented a low-pass filtered map at 8 Å as Supplementary Fig. 2d, to show the missing region. We used 8 Å for filtering because it was the best for seeing the D-loop density.

Comments:

- The figures are in general very difficult to read, since many different structures or conformations are overlaid. The authors should present morphs between the different structures in supplementary movies.

Response:

We followed the reviewer's suggestion and made Supplementary Movies 1-4, to show the transitions between the actin forms.

Comments:

- The figures are in general very 'naked' and can only be understood after careful reading of the legend. This should be changed and more details given directly in the figure using labels.

Response:

We added labels to the figures.

Comments:

- Line 137ff: The conformational changes are difficult to see in the figure. I suggest to prepare a movie showing the transition of the single actin protomer changing its conformation from F-actin (F-form) to F-actin (C-form) and then from F-actin (C-form) to G-actin (G-form).

Response:

We agree with this comment, so we generated movies showing the G-C transition and G-F transition. The F-C transition is a simple combination of the two transitions.

Comments:

- Line 187ff: The ID-ID and OD-ID inter- and intra-strand interactions should be shown

in detail – not just the backbone. Figure 3 does not help to understand this paragraph.

Response:

We replaced Fig. 3c to present side chains on the interface.

Comments:

- Line 202: B-subunit should be explained.

Response:

B-subunit is defined (lines 189-191)

Comments:

- Movies: binding instead of binding or bining

Response:

We did not notice this. We appreciate the comment.

Comments:

- Line 274ff: In their discussion, the authors refer to crystal structures of actin in the F-form. The binding mechanism of cofilin is actually completely based on these structures and derived MD simulations. However, because the manuscript or the data have been not made available to the reviewers, it is difficult to judge on this part of the discussion. For example, is the move of SD1 really as extensive as described (line 289)?

Comments:

We agree with the referee. We decided not to use unpublished data and we cited a study showing very similar computer simulation results to ours.

Comments:

- Figure 7 and 8. Actin protomers should not be shown as spheres but differently to show their polarity.

Response:

We agreed with this comment and modified Figs. 7 and 8.

Reviewer #3 (Remarks to the Author):

This paper describes the structure of actin filament fully decorated with cofilin solved by cryo-electron microscopy and image analysis. The resolution is 3.8 Å, and this is high enough for the authors to build a reliable atomic model and discuss the binding interactions between actin protomers and cofilin as well as the conformational changes of actin upon cofilin binding in detail. Based on such structural information, the authors also extend discussion on how the structural changes of actin occur, why cofilin preferably binds to ADP-F-actin, how cooperative binding occurs along the actin filament, and how a clustered binding of cofilin severs the actin filament. This is a solid work, and the manuscript is comprehensible in most places.

Some points of concern are listed below for the improvement of the manuscript.

Comments:

1. Many different aspects of the structure are nicely presented in the figures, but almost no labels of actin domains and residues make them rather difficult for general readers to understand what they see and even the direction of view of each figure.

Response:

We followed the reviewer's suggestion and added labels to the figures.

Comments:

2. The terms F-site, Gi-site (Gi_l-site, Gi_s-site) and Go-site are defined in the section of "Actin-cofilin interactions", but the reasons for these naming are not well explained. Since the F-site is on the P-subunit (actin subunit on the pointed-end side) and the G-site is on the B-subunit (actin subunit on the barbed-end side), it would be easier to follow the text if the F-site and G-site are termed P-site and B-site, respectively.

Response:

In ref 36, Mannherz et al., coined the terms F-site and G-site, so we followed these terms. We added sentences to describe this (lines 205, 223-224).

Comments:

3. "The left side of the Gi-site" on lines 224-225 is difficult to follow in Fig. 4.

Response

We changed the description to "the furthestmost inner area" (line 229).

Comments:

4. The last section of Discussion on “Effects on depolymerization rates at the ends” is a little too speculative and is better to be avoided at this stage.

Response

We agree that it includes speculation because it is just one possible model. However, the binding model for explaining the severing can explain the effects on depolymerization without modification. We stated that this is just one possible model, hence, we would like to leave the description in this section (lines 447-450).

Comments:

5. Line 137: “this rotational axis as the G/F axis”

Response

We corrected the erratum.

Comments:

6. Line 282: “the network is less extended”

Response

We deleted this description when we updated the MS.

Comments:

7. Lines 293 and 329-330: “local concentration” is an awkward wording.

Response

We changed the descriptions.

Comments:

8. Line 326: “the OD-ID interaction”

Response

We corrected the erratum.

Comments:

9. The sentence on lines 392-395 is difficult to follow.

Response

We modified this sentence.

Reviewers' comments:

Reviewer #1 (Remarks to the Author):

I remain convinced that this work is an important advance in our understanding of the interaction of cofilin with actin filaments. The quality of the structural data is high and adequate for the conclusions drawn. Resolution of the side chains in the new structure is essential for describing the interactions. I would have devoted much more of the manuscript to describing these details and much less to the speculative discussion. This would have been a strong argument against the criticism from the other reviewers about a lack of novelty.

The authors responded constructively to my suggestions by revising the figures and the text. These changes in the results section satisfy me. I still have minor concerns about some aspects of the figures noted below. Many of the conclusions are sound, but other conclusions seem to be based on a relatively narrow consideration of the available evidence. I agree with reviewer 3 that the discussion is overly long.

Fig. 3AB: These images show contacts between subunits in filaments \pm cofilin. I suggested replacing the ribbon diagram with stick figures or space-filling models or a combination of those two styles. The authors responded with stick figures for some of the side chains, which is better, but they do not illustrate the interactions as clearly as a cofilin stick figure on a space-filling actin structure. Showing these interfaces clearly is important for this paper, where the higher resolution reveals these important details better than previous work.

Fig. 3CD: I was concerned that the orientation of the images showing interstrand contacts in D and E was not helpful, since the contacts are difficult to see in such a projection image. The authors responded by changing the colors. I appreciate that illustrating these interfaces is challenging, but the revised figure is not much more informative than the original.

Text: My first review pointed out multiple aspects of the discussion, which I found to be overly speculative and/or did not take into account all of the available information. In most cases the authors stuck with their interpretation of the mechanism of action of cofilin based on their new structure rather than considering other evidence that I suggested. This is their privilege to speculate, but I predict that some aspects of their proposed mechanism will turn out to be incorrect. Furthermore, they did not take into account one of the leading hypotheses regarding the mechanism of action of cofilin, namely that it displaces a critical divalent cation. A relevant paper (28) is cited, but only in the context of disorder of the D-loop.

The text continues to mix up the effects of cofilin and ADF. Here is one example on lines 413-436: The authors wrote "cofilin can directly bind to the actin subunit at the B-end of the filament and accelerates depolymerization (ref 17)," but ADF, not cofilin, was used in the cited experiments, which were done at alkaline pH. The authors responded by changing the text to acknowledge that the work in ref 17 was done with ADF at alkaline pH, but they did not modify their conclusion in spite of the fact their excellent new work in this paper is on cofilin, not ADF.

Revised text line 65 has another example of this problem: "ADF/cofilin also accelerate depolymerization of the filament at basic pH (7.8 or 8.0)." ADF had a much stronger effect, so it is not appropriate to imply that depolymerization is a feature of both proteins by using "ADF/cofilin" in this sentence and throughout this paragraph. This lack of precision will mislead readers and take years to correct in the literature.

The authors did delete the section on treadmilling as advised, which makes the paper better.

We appreciate the comments of the reviewer 1 to improve our MS.

Reviewer #1 (Remarks to the Author):

Comments:

Fig. 3AB: These images show contacts between subunits in filaments \pm cofilin. I suggested replacing the ribbon diagram with stick figures or space-filling models or a combination of those two styles. The authors responded with stick figures for some of the side chains, which is better, but they do not illustrate the interactions as clearly as a cofilin stick figure on a space-filling actin structure. Showing these interfaces clearly is important for this paper, where the higher resolution reveals these important details better than previous work.

Response:

We made a supplementary movie 1 to show the interaction.

Comments:

Fig. 3CD: I was concerned that the orientation of the images showing interstrand contacts in D and E was not helpful, since the contacts are difficult to see in such a projection image. The authors responded by changing the colors. I appreciate that illustrating these interfaces is challenging, but the revised figure is not much more informative than the original.

Response

As the reviewer mentioned, it was difficult to illustrate the molecular interface by still images. We instead prepared supplementary movies 2 and 3 to show the interface from multiple angles.

Comments

Text: My first review pointed out multiple aspects of the discussion, which I found to be overly speculative and/or did not take into account all of the available information. In most cases the authors stuck with their interpretation of the mechanism of action of cofilin based on their new structure rather than considering other evidence that I suggested. This is their privilege to speculate, but I predict that some aspects of their proposed mechanism will turn out to be incorrect. Furthermore, they did not take into

account one of the leading hypotheses regarding the mechanism of action of cofilin, namely that it displaces a critical divalent cation. A relevant paper (28) is cited, but only in the context of disorder of the D-loop.

Response

Of course our models require further confirmations and some part of them might be wrong, as usual reports.

In Ref28, the “stiffness cation” is suggested to interact with the basal part of the D-loop and to strengthen the connections between two actin subunits in one strand. The stiffness cation dissociates from the filament during the cofilactin formation. The stiffness cation is important for actin filament stabilization and it should affect the cofilin functions indirectly.

However, it might be difficult to imagine that the stiffness cation release is the main factor for the cofilin functions. The sites for the stiffness cation binding described in Ref28 are distant from the cofilin binding sites, indicating that the initial cofilin binding to the actin filament cannot directly affect the stiffness cation. We believe that the stiffness cation will be released as a result of the structural change induced by cofilin binding. The OD-ID interaction in the strand is broken, which destabilizes the D-loop structure and makes it difficult to keep stable binding of the stiffness cation.

Comments:

The text continues to mix up the effects of cofilin and ADF. Here is one example on lines 413-436: The authors wrote "cofilin can directly bind to the actin subunit at the B-end of the filament and accelerates depolymerization (ref 17)," but ADF, not cofilin, was used in the cited experiments, which were done at alkaline pH. The authors responded by changing the text to acknowledge that the work in ref 17 was done with ADF at alkaline pH, but they did not modify their conclusion in spite of the fact their excellent new work in this paper is on cofilin, not ADF.

Response:

Wioland et al. reported that human ADF and mouse cofilin-2 has similar amount of effects on the acceleration of depolymerization of ADP-actin filaments (Fig. 3C in ref 17). Therefore, we believe our description is precise although mouse cofilin-1 is less effective. Chicken cofilin we used is relatively closer to mouse cofilin-2 than to mouse cofilin-1.

Comments:

Revised text line 65 has another example of this problem: "ADF/cofilin also accelerate depolymerization of the filament at basic pH (7.8 or 8.0)." ADF had a much stronger

effect, so it is not appropriate to imply that depolymerization is a feature of both proteins by using "ADF/cofilin" in this sentence and throughout this paragraph. This lack of precision will mislead readers and take years to correct in the literature.

Response:

In introduction, we referred to two experiments in Wioland et al., 2017 as “Depolymerization at the P-end is accelerated in an ADF/cofilin-saturated actin filament (hereafter cofilactin filament). When no ATP-G-actin is available, ADF/cofilin binds to and accelerates the dissociation of the last barbed end (B-end) subunit of a bare actin filament.”

These are precise descriptions because Wioland et al., reported these effects by using both of mouse cofilins and human ADF. For the first sentence, Wioland et al. reported that human ADF and mouse cofilin-2 has similar amount of effects on the acceleration of depolymerization of ADP-actin filaments (Fig. 3C in ref 17), as we described. For the second sentence, the depolymerization rates at the P-end of the cofilin-saturated filaments were 2-6 times faster than those of bare actin filaments (Fig. 4C in ref 17).

However we understand the reviewer’s concerns because these effects were observed under certain conditions which are not obvious whether they are physiological. We added terms “under certain conditions” after “ADF/cofilin also accelerate depolymerization of the filament at basic pH (7.8 or 8.0)” (line 65). And we deleted depolymerization effects from abstracts (line 38), the last of introduction (line 92-93) and one sentence in introduction (line 71-72).

It is also true that the length of the current MS exceeded the limitation of the format because of the long “Discussion”. We moved the main part of “Effects on depolymerization rates at the ends” section to “Supplementary Discussion”. We also moved Fig. 8 to Supplementary Fig. 6.